# An optimized CRISPR/Cas9 approach for precise genome editing in neurons

**Huaqiang Fang[1,2,3,4†‡§], Alexei M Bygrave[1†], Richard H Roth[1], Richard C Johnson[1], Richard L Huganir[1,2]***

[1]Department of Neuroscience, Johns Hopkins University School of Medicine, Baltimore, United States; [2]Kavli Neuroscience Discovery Institute, Johns Hopkins University, Baltimore, United States; [3]PKU-Nanjing Institute of Translational Medicine, Nanjing, China; [4]Research Unit of Mitochondria in Brain Diseases, Chinese Academy of Medical Sciences, Nanjing, China

**Abstract** The efficient knock-in of large DNA fragments to label endogenous proteins remains especially challenging in non-dividing cells such as neurons. We developed *Targeted Knock-In* with *Two* (TKIT) guides as a novel CRISPR/Cas9 based approach for efficient, and precise, genomic knock-in. Through targeting non-coding regions TKIT is resistant to INDEL mutations. We demonstrate TKIT labeling of endogenous synaptic proteins with various tags, with efficiencies up to 42% in mouse primary cultured neurons. Utilizing in utero electroporation or viral injections in mice TKIT can label AMPAR subunits with Super Ecliptic pHluorin, enabling visualization of endogenous AMPARs in vivo using two-photon microscopy. We further use TKIT to assess the mobility of endogenous AMPARs using fluorescence recovery after photobleaching. Finally, we show that TKIT can be used to tag AMPARs in rat neurons, demonstrating precise genome editing in another model organism and highlighting the broad potential of TKIT as a method to visualize endogenous proteins.

**\*For correspondence:**
rhuganir@jhmi.edu

[†]These authors contributed equally to this work

**Present address:** [‡]PKU-Nanjing Institute of Translational Medicine, Nanjing, China; [§]Research Unit of Mitochondria in Brain Diseases, Chinese Academy of Medical Sciences, Nanjing, China

**Competing interests:** The authors declare that no competing interests exist.

## Introduction

To learn about protein function in complex systems biologists need tools to visualize endogenous proteins. A powerful approach to achieve this is the introduction of genetic tags to proteins of interest, fluorescent or otherwise. Endogenous protein labeling is highly advantageous compared to other approaches. First, it removes the dependence on specific antibodies which require extensive validation. Second, introducing fluorescent tags circumvents difficulties of antibodies physically accessing their targets, such as core components of the postsynaptic density (PSD), in which the antibody epitopes can be sterically occluded by other proteins. Third, genetic tagging with fluorescent proteins is compatible with longitudinal imaging in live tissue in vitro and in vivo. Finally, tagging endogenous proteins prevents potential confounds of protein mis-localization following overexpression and preserves the natural transcriptional and posttranscriptional regulation of the gene. The discovery of CRISPR/Cas9 changed the gene editing field (*Cho et al., 2013*; *Cong et al., 2013*; *Doudna and Charpentier, 2014*; *Hsu et al., 2014*; *Jinek et al., 2012*; *Mali et al., 2013*; *Sander and Joung, 2014*), making genome editing much easier, faster, and cheaper. Transgenic knock-in organisms can be generated to tag specific proteins in every cell. CRISPR/Cas9 technology has facilitated this approach; however, it is still relatively time-consuming and considered low throughput, especially in rodents. In addition, having knock-in (KI) in all cells in an organism can be problematic in dense tissues. Therefore, methods to enable genome editing in single or a relatively sparse population of cells are highly desirable. Such CRISPR/Cas9-based genome editing strategies traditionally rely on homology-dependent recombination (HDR) (*Lombardo et al., 2011*; *Wang et al., 2013*; *Yang et al., 2013*); however, HDR does not work efficiently in post-mitotic cells,

such as neurons (*Mao et al., 2008*; *Nishiyama et al., 2017*; *Orthwein et al., 2015*; *Saleh-Gohari and Helleday, 2004*). Consequently, in post-mitotic cells researchers have adopted methods based on non-homologous end joining (NHEJ) (*Bétermier et al., 2014*; *Cox et al., 2015*; *Gao et al., 2019*; *Lieber, 2010*; *Suzuki et al., 2016*; *Willems et al., 2020*). Such approaches have achieved higher KI efficiency in neurons compared to HDR-based methods. However, these approaches directly edit the coding sequence of the target gene, possibly causing (insertion or deletion) INDEL mutations, which commonly occur at CRISPR/Cas9 mediated editing sites (*Suzuki et al., 2016*). This is especially a concern if the tag is not introduced at the C-terminal portion of the protein. Furthermore, even when the KI proceeds as planned, the deletion or addition of some amino acids flanking the tag is unavoidable, and the exact site of KI is still dependent on a (protospacer-adjacent motif) PAM and reliable guide to be present at the right place.

To address these problems, we designed a new strategy – *T*argeted *K*nock-*I*n with *T*wo (TKIT) guides – to enable precise genome editing by utilizing two guide RNAs located in non-coding regions upstream and downstream of the coding sequence to be edited. This strategy protects from INDEL mutations within the coding region and enables absolute control over the sequence surrounding the KI site. In addition, since the coding region is not being targeted directly, we have fewer limitations during the selection of potential guide RNAs, increasing the availability of high-score guides and lowering the likelihood of off-target editing.

In this study, we have used TKIT to KI Super Ecliptic pHluorin (SEP), a pH dependent GFP variant (*Lin and Huganir, 2007*; *Miesenböck et al., 1998*), and other protein tags to the N-terminus of α-amino-3-hydroxy-5-methyl-4-isoxazolepropionic acid (AMPA) and N-methyl D-aspartic acid (NMDA) receptor subunits, which form glutamate receptors responsible for the majority of excitatory transmission in the brain. We show that TKIT enables efficient KI in primary cultured neurons from mouse and rat and that TKIT can be used for genome editing in vivo with in utero electroporation or through injection of adeno-associated viral (AAV) vectors in adult mice. Thus, TKIT provides an efficient, easy to use, and readily scalable approach for the tagging of endogenous proteins to facilitate their study in intact biological systems.

## Results

Homology Independent Target Integration (HITI) based KI strategies have been used to tag endogenous proteins in postmitotic cells such as neurons (*Gao et al., 2019*; *Suzuki et al., 2016*; *Willems et al., 2020*); however, these approaches are sensitive to INDEL mutations as the coding sequence is directly edited. This is particularly problematic when labeling the N-terminal region of proteins, where the introduction of INDELs will be most damaging. Additionally, in the vast majority of cases, a Cas9 cutting site is not present at exactly the desired site of insertion. Therefore, the tag cannot always be precisely targeted, and deletion or addition of extra base pairs is unavoidable even when editing occurs as planned and without generation of INDELs. Further, the choice of guides with high cutting efficiency and good off-target scores near the desired insertion site is often very limited. In an attempt to overcome these issues – which are common to all currently used NHEJ-based CRISPR KI methods – we designed a modified CRISPR/Cas9-based KI strategy. Our approach utilizes two guide RNAs that cut the genomic DNA flanking the coding region of interest and includes a replacement donor DNA fragment encoding the excised coding sequence with the addition of a fluorescent or small (i.e., Myc) protein tag. We named this new CRISPR/Cas9 based knock-in strategy TKIT guides.

### Genome editing of coding sequence by targeting non-coding regions

As a proof of principle, we sought to KI SEP, a pH sensitive GFP variant that is used to visualize cell surface proteins, to the N-terminus of the AMPA receptor subunit GluA2, encoded by the *Gria2* gene. We designed guides to cut within the 5'-UTR (untranslated region) and intron 1–2 of *Gria2*, choosing guides located ~100 bp away from splice junctions so as not to affect mRNA processing (*Figure 1A*). We constructed a donor DNA fragment containing the endogenous *Gria2* sequence, with the addition of a SEP-tag behind the signal peptide, and the same two guides and PAM sequences at the ends. The two guides within the donor have opposite locations and flipped sequences compared to the genomic DNA (*Figure 1A*, *Figure 1—figure supplement 1*). We added this switch-and-flip design to promote forward insertion of the donor DNA, which occurs randomly

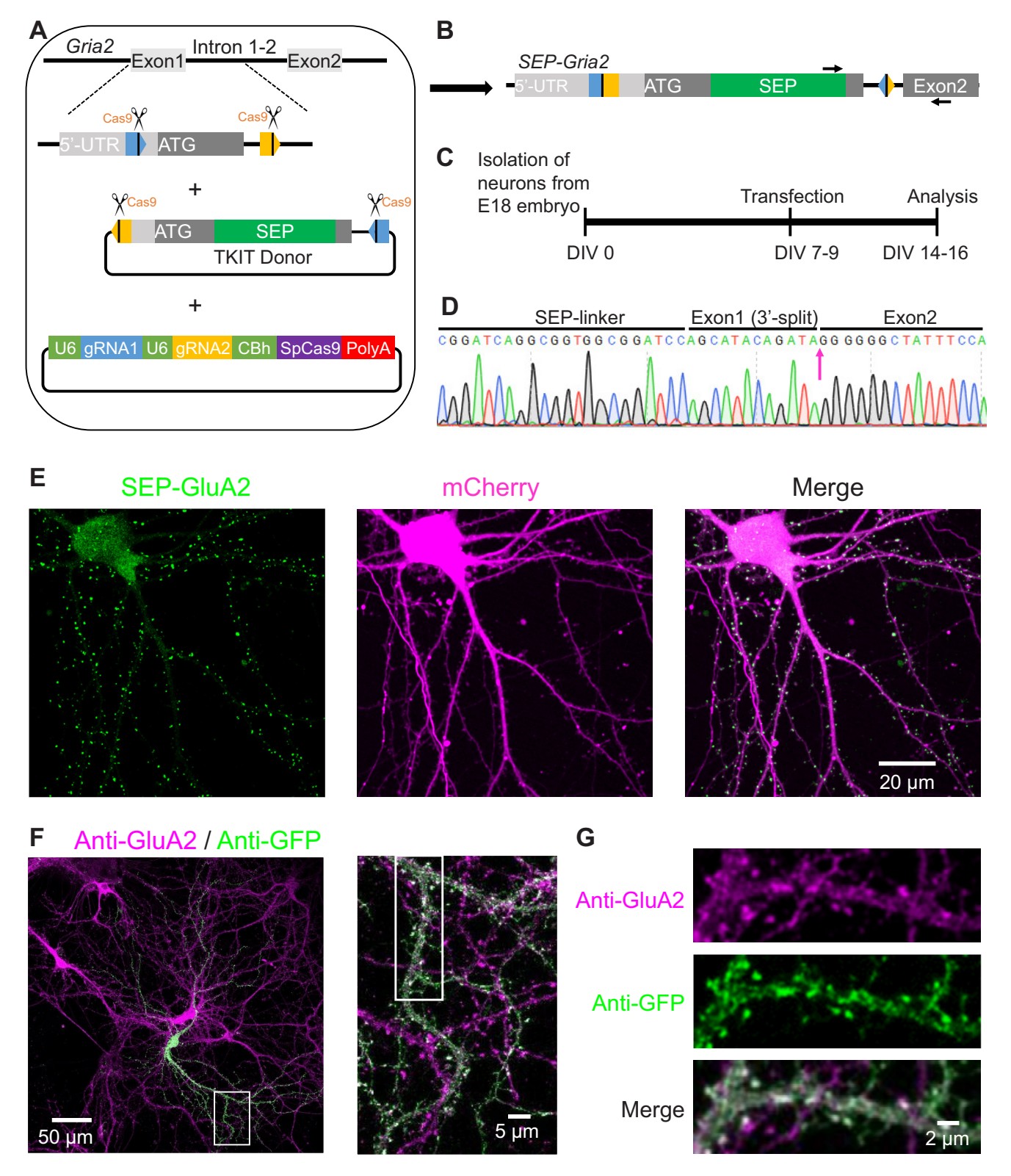

**Figure 1.** TKIT-mediated genome editing in neuron culture. (**A**) Graphical representation of the mouse genomic loci of *Gria2* and schematic of TKIT gene targeting. The blue and orange pentagons indicate gRNA target sequences. The black vertical line within pentagon indicates the Cas9 cleavage site. (**B**) Schematic of genomic DNA following SEP KI to *Gria2* with TKIT. Arrows indicate primers for RT-PCR and cDNA sequencing. (**C**) Experimental scheme for SEP KI in primary cultured neurons. (**D**) Sequence analysis of the cDNA from mRNA extracted from electroplated neuronal culture of SEP-

Figure 1 continued

GluA2. (E) Representative images showing the endogenous SEP signal in live neurons following TKIT-mediated SEP-GluA2 knock-in. Scale bar, 20 µm. (F) Representative images of immunofluorescence showing that SEP-GluA2 (green) colocalized with GluA2 (magenta). Scale bars display 50 or 5 µm. (G) Enlarged dendrite and spine (white boxes in right panel of F). Scale bars display 2 µm.

The online version of this article includes the following figure supplement(s) for figure 1:

**Figure supplement 1.** Strategy to promote donor DNA insertion in the correct orientation.

through the process of NHEJ. If the donor inserts into the genome in the reverse orientation (and without INDELs) the guides and PAMs remain intact so they will be cut again by Cas9, enabling another chance at insertion in the correct orientation. The cycle will continue until the donor is inserted in the correct orientation, or until INDELs destroy the guides and PAM sequence (*Figure 1B*, *Figure 1—figure supplement 1*).

To test TKIT we transfected primary mouse cortical cultures at DIV7-9 with plasmids containing: (1) both guide RNAs along with SpCas9, (2) SEP-GluA2 donor DNA fragment, and (3) mCherry as a cell morphology marker (*Figure 1A,C*). At DIV14-16 we live-imaged transfected neurons and found neurons with punctate SEP signal concentrated in dendritic spines, consistent with the labeling of AMPARs (*Figure 1E*). We used immunofluorescence with antibodies against GFP and the C-terminus of GluA2 to show that the GFP signal was co-localized with GluA2 (*Figure 1F–G*).

To confirm that SEP KI to the *Gria2* gene still results in normal RNA splicing, we extracted bulk mRNA from DIV19 neurons electroporated with TKIT components, conducted reverse transcription to generate cDNA, and performed PCR and Sanger sequencing. Sequencing results showed that the splice junctions between exon1 and exon2 were intact, indicating that SEP KI did not affect *Gria2* mRNA splicing (*Figure 1D*, *Figure 2—figure supplement 1*). Immunostaining of GluA2 in primary cultured neurons with TKIT mediated SEP-GluA2 KI showed that the expression level of SEP-GluA2 is comparable with GluA2 in non-transfected surrounding neurons (*Figure 2—figure supplement 1*). KI was dependent on co-transfection of the TKIT donor and the Cas9 with two guides constructs, as transfection of the TKIT donor alone did not lead to any detectable KI (*Figure 2—figure supplement 1*). These data indicate that TKIT enables accurate genomic editing in neurons, with broad potential for future applications in neurons and other post-mitotic cells.

## Efficient targeting of synaptic proteins with TKIT

NMDA and AMPA receptors mediate the majority of excitatory synaptic transmission in the mammalian CNS and are critical for the tuning of synaptic strength. In particular, the trafficking of AMPA receptors is known to underlie synaptic plasticity, supporting learning, and memory (*Diering and Huganir, 2018*). To date, the vast majority of AMPA receptor imaging studies have relied on the overexpression of (often SEP) tagged receptors (*Ashby et al., 2006*; *Kopec et al., 2006*; *Lin and Huganir, 2007*; *Thomas et al., 2008*). While these studies have helped us learn about the regulation of AMPA receptors in vitro, and more recently, in vivo (*Roth et al., 2020*; *Tan et al., 2020*; *Zhang et al., 2015*), there is always the concern that overexpression could result in aberrant and non-physiological receptor regulation. Therefore, tools to tag endogenous AMPARs will be highly useful.

We sought to tag a variety of synaptic proteins, including AMPA and NMDA receptor subunits, extending our proof of principle experiments showing SEP KI to GluA2. We designed TKIT constructs for SEP KI to the N-terminus of AMPAR subunits GluA1, GluA2, and GluA3, as well as the most abundant NMDAR subunits (NR1, NR2A, and NR2B) and the beta2 subunit of the gamma-aminobutyric acid (GABA) receptor. In addition, we wanted to demonstrate that TKIT also works for cytosolic proteins. To do this we targeted gephyrin, a postsynaptic scaffold protein that is localized to inhibitory synapses, as a candidate for GFP KI. Since previous reports tagged the N-terminus of gephyrin (*Lardi-Studler et al., 2007*; *Tyagarajan et al., 2011*), we also targeted GFP KI to the N-terminus of gephyrin. To target this set of synaptic proteins, guide cutting sites were located either in introns flanking the target exon, or in the 5'UTR of the targeted exon and the downstream intron (*Figure 2—figure supplement 2*).

As before, primary mouse cultured neurons were transfected with TKIT constructs at DIV7-9, and confocal images were captured between DIV14 and 16 following immunofluorescence staining. The

SEP-GluA1-3 signal in neurons, visualized with GFP immunostaining, showed clear puncta at dendritic spines (*Figure 2A–C*). As expected, GFP-gephyrin (*Figure 2D*) and SEP-Beta2 (*Figure 2—figure supplement 3*) showed punctate signals along dendrites.

To calculate the efficiency of TKIT KI we counted the number of KI cells as a percentage of all transfected neurons based on the mCherry cell fill. The efficiency for SEP KI to AMPA receptor subunits GluA1, GluA2, and GluA3 was 21.9%, 38.9%, and 20.5%, respectively (*Figure 2E–G*). The efficiency for Beta2 KI was 27.2% (*Figure 2—figure supplement 3*) and 41.7% for GFP-gephyrin KI (*Figure 2H*). Targeting NMDA receptors resulted in similar spine enrichment for SEP-NR1, SEP-NR2A, and SEP-NR2B (*Figure 3A–C*) with efficiencies of 15.2%, 16.7%, and 27.5%, respectively (*Figure 3D–F*).

In addition to SEP, we used TKIT to KI the larger fluorescent protein tdTomato to Gephyrin (*Figure 2—figure supplement 4*) and GluA2 (*Figure 2—figure supplement 5*) as well as the smaller tag Myc to GluA1 and GluA2 (*Figure 2—figure supplement 5*). We validated the fidelity of KI by showing colocalization of GFP and the endogenous proteins for SEP-GluA1, SEP-GluA2, and SEP-GluA3 (*Figure 2—figure supplement 6*), and SEP-NR1, SEP-NR2A, and SEP-NR2B (*Figure 3—figure supplement 1*) with immunofluorescence. Correlation analysis showed strong correlation between the intensities of anti-GFP signal and anti-GluA1-3 signal in TKIT-mediated SEP KI neurons (*Figure 2—figure supplement 6*). We further validated the correct subcellular localization of proteins tagged with TKIT. We verified the synaptic localization of SEP-GluA1-3 with immunofluorescence using antibodies against GFP and PSD95 (*Figure 4A–C*).

We observed a strong correlation between the anti-GFP signal and anti-PSD95 signal in SEP KI neurons (*Figure 4D–F*). We found that GFP- or tdTomato-gephyrin signal colocalized with gephyrin and the GABA receptor Beta2 subunit, used as a marker for inhibitory synapses (*Figure 2—figure supplement 4*). Sequence analysis of reverse-transcribed cDNA from mRNA extracted from neuronal cultures electroporated with each TKIT construct showed intact splicing of all tagged proteins in this study (*Figure 4—figure supplement 1*). TKIT-mediated KI of various tags to endogenous proteins will aid the characterization and functional study of proteins avoiding potential artifacts of protein overexpression.

Because guides flank a critical portion of the gene being edited, it is possible that a portion of exon 1 is excised and repaired without the addition of the donor DNA fragment. This would lead to the generation of a knock-out (KO) allele for the target gene. To assess the KI/wild-type (WT)/KO possibility in the *Gria2* KI experiments in neuronal cultures described in *Figure 1*, we designed primer-pairs flanking the two guides KI site and performed genomic PCR. Genomic DNA samples were extracted from neuron culture electroporated with TKIT constructs. The primer-pair can pick up three fragments (KI/WT/KO bands). DNA electrophoresis of the PCR products showed clear KI and WT bands, but no detectable KO band (*Figure 4—figure supplement 2*). In addition, we transfected primary neuron cultures with SpCas9 and two guides, SEP-GluA2 donor DNA and mCherry as a morphology marker at DIV 7–9 and then used immunofluorescence to detect GFP and GluA2 at DIV 14–16. GluA2 was not detectable in 12.4% of the mCherry-positive cells, indicating knock-out of GluA2 in a small subset of cells that failed to knock-in the donor DNA (*Figure 4—figure supplement 2*). To test the possibility that a subset of neurons naturally lacked GluA2 expression, independent of manipulation, we assessed the levels of GluA2 in mouse cortical neurons. We found that 1.2% of cells had undetectable GluA2 (*Figure 4—figure supplement 3*). Therefore, we think that undetectable GluA2 observed in some cells transfected with TKIT plasmids is an unfortunate, but not completely unexpected, result of gene knock-out through double strand break generation without donor insertion.

## Endogenous KI of SEP-GluA2 in vivo

After demonstrating efficient KI in primary cultured neurons, we sought to test the utility of TKIT to introduce a SEP-tag to the N-terminus of GluA2 in vivo. Plasmids containing SEP-GluA2 donor DNA and complementary two guides with spCas9 were transfected into layer 2/3 pyramidal neurons of somatosensory cortex using in utero electroporation of mouse embryos at embryonic day 15 (E15; *Figure 5A*). Cranial windows were then implanted over the barrel cortex of adult animals. After 2–3 weeks of recovery in vivo two-photon images were acquired (*Figure 5B*). Punctate SEP-GluA2 could be clearly observed along the dendrites of layer 2/3 pyramidal neurons (*Figure 5C and D*, *Figure 5—video 1*), similar to previously studies showing the over-expressed SEP-GluA1 and SEP-GluA2

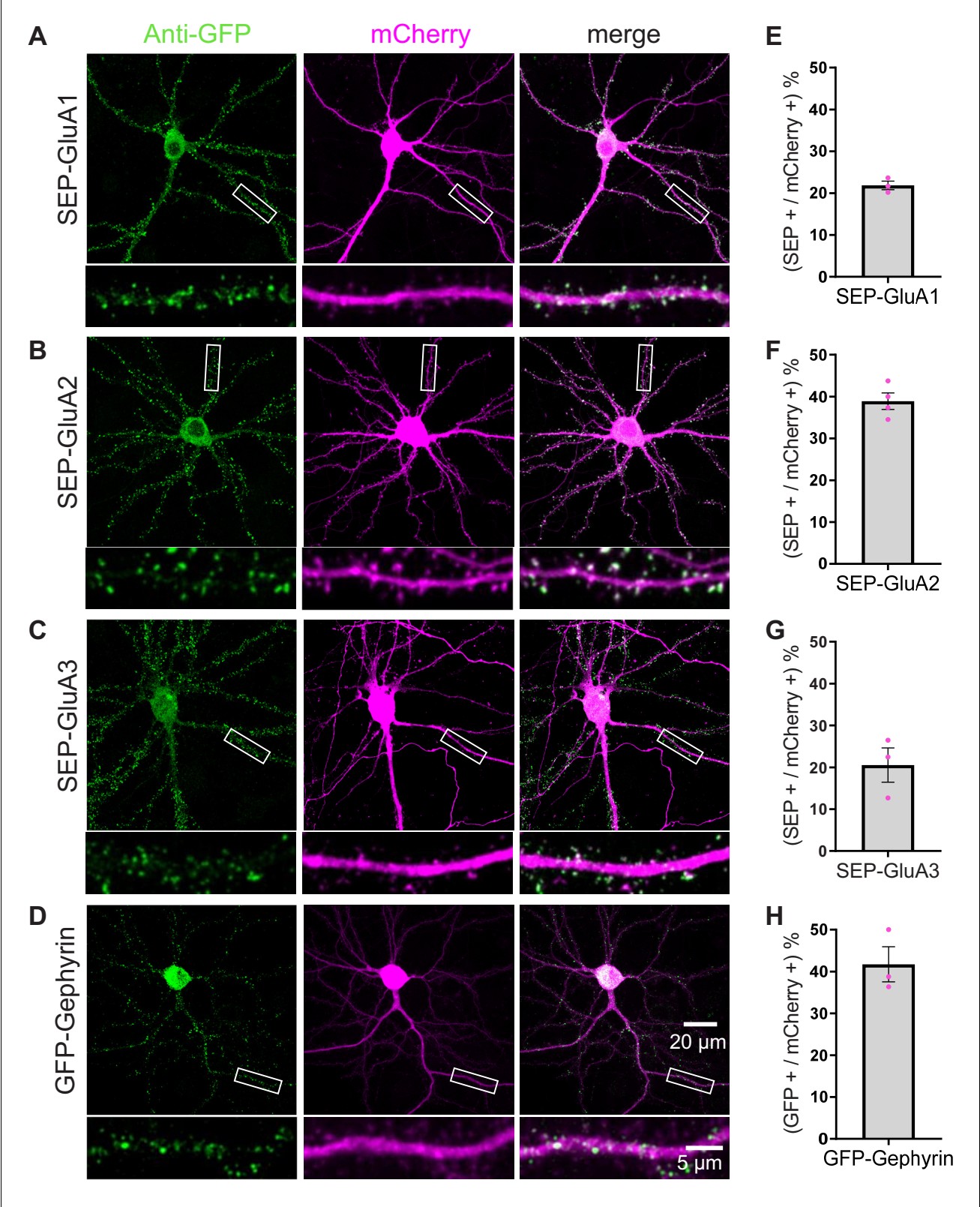

**Figure 2.** Efficient KI of SEP-GluA1-3 and GFP-gephyrin using TKIT. (**A**) SEP-GluA1, (**B**) SEP-GluA2, (**C**) SEP-GluA3, and (**D**) GFP-gephyrin visualized with confocal microscopy after neurons were co-transfected with TKIT constructs and a mCherry cell fill. Segments of dendrite in the white boxed region are enlarged below. Scale bar, 20 μm or 5 μm. Average efficiency of KI for (**E**) SEP-GluA1, (**F**) SEP-GluA2, (**G**) SEP-GluA3, and (**H**) GFP-gephyrin. Error bars display SEM.

*Figure 2 continued on next page*

*Figure 2 continued*

The online version of this article includes the following source data and figure supplement(s) for figure 2:

**Source data 1.** Efficiency counts to assess TKIT knockin of SEP-GluA1, SEP-GluA2, SEP-GluA3, and GFP-Gephyrin in mouse neurons.

**Figure supplement 1.** Correct splicing of SEP-GluA2 and comparable GluA2 levels in KI and WT surrounding neurons.

**Figure supplement 1—source data 1.** GluA2 expression levels in SEP-GluA2 knockin and untransfected neurons.

**Figure supplement 1—source data 2.** SEP-GluA2 knockin efficiency compared to donor-only control.

**Figure supplement 2.** Location of TKIT guide pairs.

**Figure supplement 3.** TKIT of SEP-GABA$_A$R β2.

**Figure supplement 3—source data 1.** Efficiency counts to assess TKIT knockin of SEP-GABAAR β2 in mouse neurons.

**Figure supplement 4.** TKIT of GFP-Gephyrin and tdTomato-Gephyrin.

**Figure supplement 4—source data 1.** Immunofluorescence data for correlation of GFP-Gephyrin knockin signal with gephyrin.

**Figure supplement 5.** TKIT of alternative tags for AMPA receptors.

**Figure supplement 6.** Correlation of SEP-GluA1-3 KI with endogenous AMPA receptors.

**Figure supplement 6—source data 1.** Immunofluorescence data for correlation of SEP-GluA1, SEP-GluA2, and SEP-GluA3 knockin signal with GluA1, GluA2, and GluA3, respectively.

(*Suresh and Dunaevsky, 2017*; *Zhang et al., 2015*). We were even able to observe SEP signal in mice that were more than 1-year old, indicating the long-term stability of the knock-in tag. Therefore, using TKIT we have been able to tag, and subsequently visualize, endogenous AMPAR subunits in vivo. This approach will allow the longitudinal tracking of endogenous AMPAR plasticity in behaving animals.

## Virus-mediated TKIT in vivo

To increase the ease and flexibility in which TKIT can be applied for KI studies in vivo, we designed AAV vectors for TKIT KI. We developed a two-virus system to enable genomic editing of wild-type mice. We packaged HA-spCas9 into AAV and designed and produced AAVs to deliver two guides and donor DNA fragments within the same construct for SEP KI of GluA1 or GluA2 (TKIT-SEP-GluA1, TKIT-SEP-GluA2) (*Figure 5E*). We injected a mix of high titer AAV-spCas9 and AAV-TKIT-SEP-GluA2 into the dorsal hippocampus of adult C57BL/6 mice and allowed 3 weeks for AAV transduction, guide expression, genome editing/KI, and turnover/replacement of endogenous protein. Native SEP fluorescence signal could be clearly detected in the injected hemisphere of the hippocampus (*Figure 5F*). To calculate the efficiency of KI we used immunohistochemistry against HA to detect Cas9, and GFP to detect KI cells (*Figure 5G*). Of the HA-positive cells we found that 16.1% were also KI positive (*Figure 5H*), indicating high efficiency of genome editing in vivo.

## Cell-type specific TKIT in vivo

To gain cell-type specificity of TKIT KI we bred heterozygous *Camk2a*^Cre^ or homozygous *Pvalb*^Cre^ mice with homozygous lox-stop-lox conditional Cas9 animals to generate offspring with Cas9 expression in forebrain *CamK2a* positive neurons or *Pvalb* (PV) positive interneurons (*Camk2a*^Cre/wt^: Cas9^fl/wt^ or *Pvalb*^Cre/wt^:Cas9^fl/wt^). We injected high titer AAV-TKIT-SEP-GluA1 into the dorsal hippocampus of adult *Camk2a*^Cre/wt^:Cas9^fl/wt^ and *Pvalb*^Cre/wt^:Cas9^fl/wt^ animals (*Figure 6A,C*). After 2–3 weeks we observed robust KI of SEP-GluA1 in the dorsal hippocampus of the injected hemisphere. Super-resolution confocal (Airyscan, see Materials and methods) images were acquired of CA1 hippocampal neurons (*Figure 6B,D*). Here, endogenous SEP signal could be seen concentrated in putative dendritic spines of excitatory neurons (*Camk2a*^Cre/wt^:Cas9^fl/wt^ animals) and along the smooth dendrites of PV neurons (*Pvalb*^Cre/wt^:Cas9^fl/wt^ animals).

Next, we tested if the TKIT-SEP-GluA2 virus could be coupled with two-photon imaging in vivo, as we demonstrated previously with in utero electroporation. To test this, the somatosensory cortex of *Camk2a*^Cre/wt^:Cas9^fl/wt^ mice was injected with high titer TKIT-SEP-GluA2 AAV. Immediately after virus injection a cranial window was placed above the injection site (*Figure 6E*). After 2 weeks, animals were imaged with two-photon microscopy. As with in utero electroporation, we could clearly observe cells with SEP-GluA2 KI in vivo (*Figure 6F*). Given the higher infection rate of TKIT-SEP-GluA2 AAV compared to in utero electroporation, we were able to visualize a higher density of SEP-GluA2 expressing neurons which will allow the longitudinal study of synaptic surface GluA2 on multiple neurons within one field of view, with KI sparse enough to distinguish individual synaptic puncta

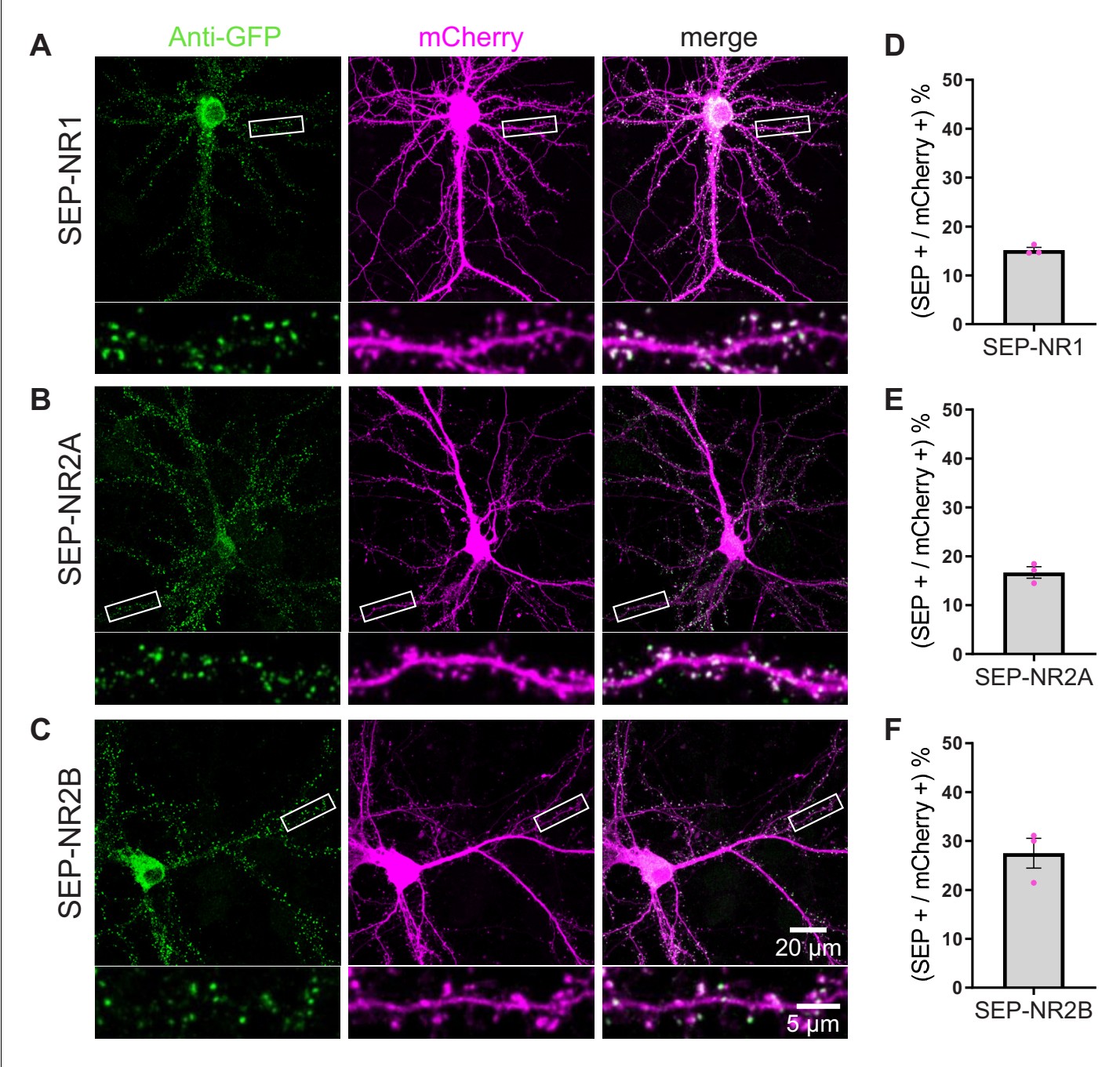

**Figure 3.** TKIT of SEP NMDA receptor subunits. (**A**) SEP-NR1, (**B**) SEP-NR2A, and (**C**) SEP-NR2B KI were visualized with confocal microscopy after neurons were co-transfected with TKIT constructs and a mCherry cell fill. Segments of dendrite in the white boxed region are enlarged below. Scale bar, 20 μm or 5 μm. Average efficiency of KI for (**D**) SEP-NR1, (**E**) SEP-NR2A, and (**F**) SEP-NR2B. Error bars display SEM.

The online version of this article includes the following source data and figure supplement(s) for figure 3:

**Source data 1.** Efficiency counts to assess TKIT knockin of SEP-NR1, SEP-NR2A, and SEP-NR2B in mouse neurons.

**Figure supplement 1.** TKIT of SEP-NMDA receptors.

and neuronal processes. These data demonstrate the utility of TKIT to KI fluorescent tags to endogenous proteins in a cell-type specific manner in adult mice following injection of a single AAV, enabling their subsequent visualization in vivo. Recent reports show that fragments of AAVs can integrate into double stranded DNA breaks induced by CRISPR/Cas9 (*Hanlon et al., 2019*;

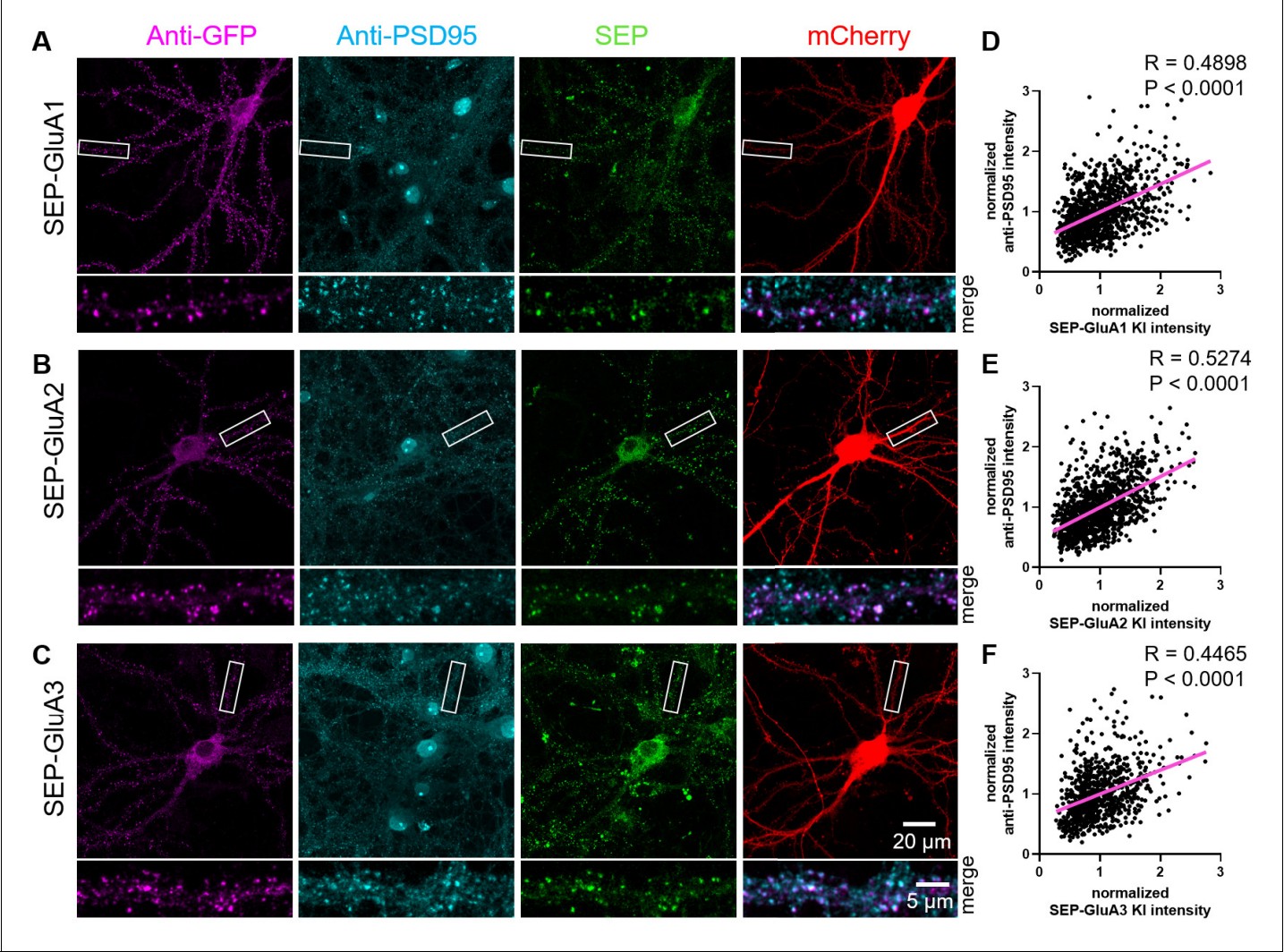

**Figure 4.** Validation of synaptic localization following SEP KI to AMPARs. TKIT of (**A**) SEP-GluA1, (**B**) SEP-GluA2, and (**C**) SEP-GluA3 with endogenous SEP signal (green), PSD95 staining (cyan), GFP staining (magenta), and mCherry cell fill (red). Segments of dendrite in the white boxed region are enlarged below. Note: the endogenous SEP signal and cell-fill channel (mCherry) were removed from the enlarged overlay images for clarity. Scale bars display 20 or 5 μm. (**D–F**) Scatter plots showing the correlation between SEP-GluA1-3 signal and PSD95 immunofluorescence signal. Correlation (slope) between SEP and anti-PSD95 fluorescent intensity was calculated by Simple Linear Regression (GraphPad Prism 7 or 8). (**D**) SEP-GluA1 correlations with data from 22 neurons and 1114 synapses, (**E**) SEP-GluA2 correlations with data from 23 neurons and 1191 synapses, and (**F**) SEP-GluA3 correlations with data from 16 neurons and 810 synapses.

The online version of this article includes the following source data and figure supplement(s) for figure 4:

**Source data 1.** Immunofluorescence data for correlation of PSD95 signal with SEP-GluA1, SEP-GluA2, and SEP-GluA3.

**Figure supplement 1.** Intact splicing with TKIT.

**Figure supplement 2.** Low levels of GluA2 KO with TKIT.

**Figure supplement 2—source data 1.** GluA2 immunofluorescence data to assess the proportion of neurons that are wild type (unedited), knockin (SEP-GluA2), or knockout for GluA2.

**Figure supplement 3.** Occasional detection of GluA2-lacking neurons in the absence of TKIT construct transfection.

**Figure supplement 3—source data 1.** GluA2 immunofluorescence data to assess the fraction of neurons that are GluA2-lacking under basal conditions.

*Wolter et al., 2020*). Presently, we do not have data to confirm if this is occurring in our experiments with TKIT-mediated KI of SEP-GluA1 and SEP-GluA2 with AAVs. Integration of AAV fragments into the 5'UTR of GluA1 or GluA2 could disrupt function, potentially contributing to the decreased efficiency observed in vivo (*Figure 5H*) compared to in vitro (*Figure 2F*).

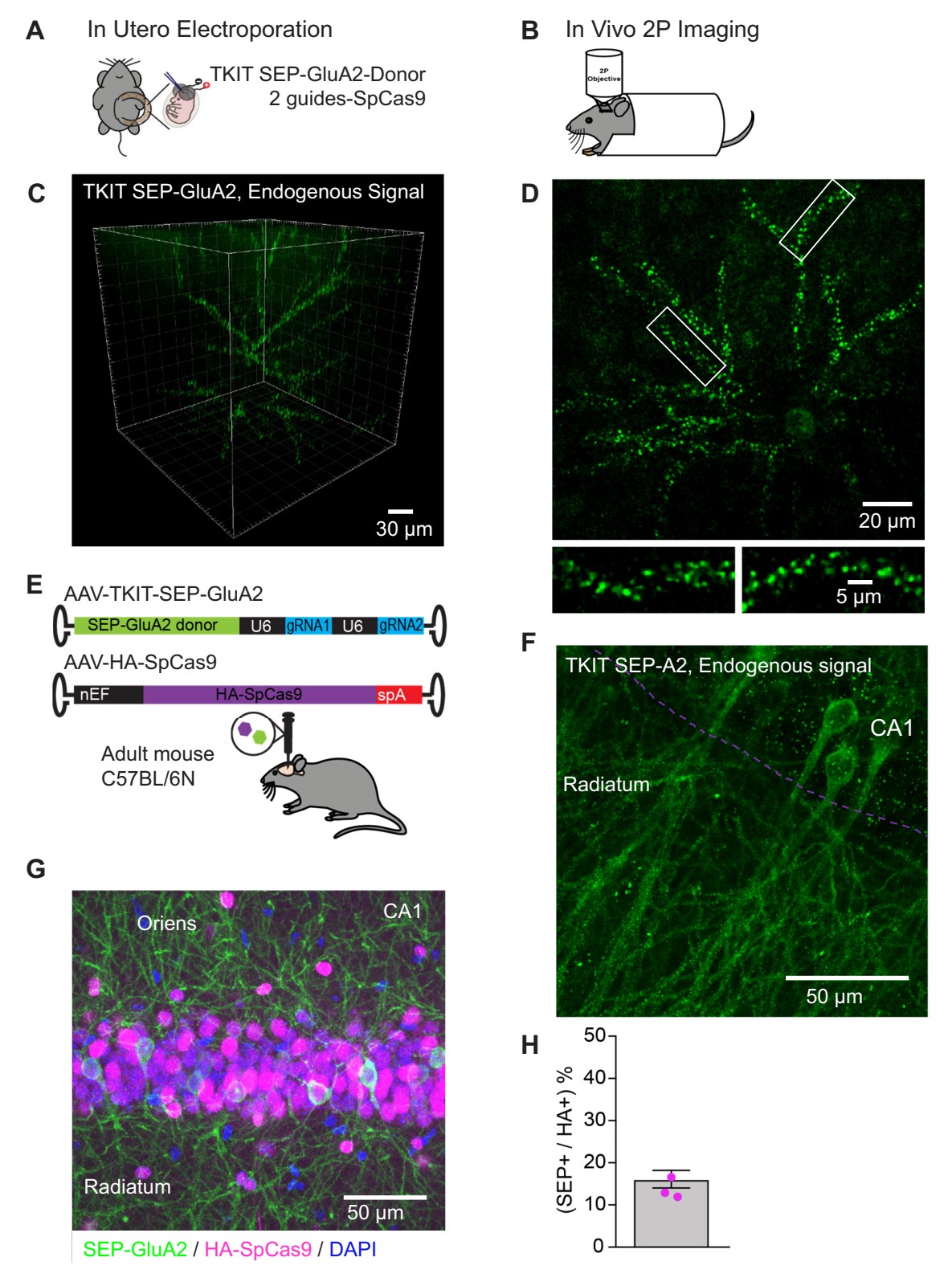

**Figure 5.** TKIT mediated KI in vivo. (**A**) Schematic of in utero electroporation and (**B**) subsequent in vivo two-photon microscopy setup. (**C**) Two-photon 3D reconstruction of a layer 2/3 pyramidal neuron with TKIT-mediated SEP-GluA2 KI. (**D**) Maximum intensity projection of the same example cell in (**C**) with magnified dendritic segments (white boxes). Scale bars are shown in the bottom right corner of each image. (**E**) Dual AAV-HA-SpCas9 and AAV-TKIT-SEP-GluA2 injections were made into the dorsal hippocampus of adult C57BL/6N mice. (**F**) Confocal image of native SEP-GluA2 fluorescence

*Figure 5 continued on next page*

*Figure 5 continued*

signal. The CA1 cell layer is illustrated with a magenta dotted line with dendrites in Stratum radiatum. The image is a maximum intensity projection. (G) A composite confocal image showing the expression of SEP-GluA2 (GFP antibody, green), Cas9 (HA antibody, magenta), and nuclei (DAPI, blue). The image is a maximum intensity projection and is centered on the CA1 pyramidal cell layer. (H) Quantification of the 16.1% of HA-SpCas9 expressing cells that are also GFP positive. Scale bars are indicated in the bottom right corner of images.

The online version of this article includes the following video and source data for figure 5:

**Source data 1.** Immunohistochemistry data for SEP-GluA2 knockin efficiency calculation with dual virus approach.

**Figure 5—video 1.** TKIT mediated SEP-GluA2 KI in vivo, punctate signal could be clearly observed along the dendrites of layer 2/3 pyramidal neurons.
https://elifesciences.org/articles/65202#fig5video1

## TKIT of SEP AMPA receptor subunits in rat neurons

After successfully using TKIT in mouse neurons in vitro and in vivo, we wanted to confirm that this approach will generalize to other species. Rats are commonly used in biomedical research, yet there are far fewer transgenic rat models compared to mice. We re-designed TKIT guides and donors to be complementary to the rat genome, to enable tagging of rat GluA1 and GluA2 with SEP. We transfected these TKIT constructs into rat hippocampal neuron culture at DIV7-8 and used GFP immunohistochemistry to detect KI cells at DIV14-15 (*Figure 7A,B*, *Figure 7—figure supplement 1*). As expected, we observed green puncta at dendritic spines, which co-localized with the synaptic marker PSD95 (*Figure 7—figure supplement 1*). This demonstrates that TKIT can also mediate editing of the rat genome, with efficiencies roughly equivalent to those observed in mouse neurons (*Figure 7C,D*).

## Monitoring trafficking of endogenous GluA1-containing AMPARs

Fluorescence recovery after photobleaching (FRAP) is a frequently used technique to assess the trafficking and mobility of proteins (*Ashby et al., 2006*; *Roth et al., 2017*). Using TKIT to KI a SEP-tag to GluA1, we tested the mobility of endogenously tagged GluA1 in primary cultured neurons. We live-imaged SEP-GluA1 KI cells with confocal microscopy, and, after acquiring baseline data, photobleached the SEP signal at a subset of dendritic spines. As expected for receptors that are constantly undergoing recycling to and from the synapse (*Arendt et al., 2010*; *Frischknecht et al., 2009*; *Heine et al., 2008*; *Kerr and Blanpied, 2012*; *Makino and Malinow, 2009*), we observed a recovery of the SEP-GluA1 signal over time (*Figure 7E*). Pooling the FRAP data from many spines (46 spines from three cells and three coverslips) enabled us to estimate the time constant for fluorescence recovery to be $\tau = 3.057$ min, and the immobile fraction of receptors to be 52.44% under baseline conditions (*Figure 7F*). These findings are comparable to some (*Heine et al., 2008*; *Kerr and Blanpied, 2012*), but not all (*Makino and Malinow, 2009*), previous reports utilizing overexpression of SEP-GluA1. We speculate that this is critically dependent on the degree of overexpression, with very high levels of overexpression increasing the mobile pool of AMPARs beyond what is observed under more physiological conditions. These data highlight the utility of TKIT to probe the endogenous regulation of proteins in intact biological systems.

## Discussion

Strategies to efficiently, and reliably, edit genomic DNA in post mitotic cells, such as neurons, are lacking behind methods based on HDR that are routinely applied to dividing cells (*Wang et al., 2013*; *Yang et al., 2013*). Existing methods, based on NHEJ, have shown promise for enabling KI of tags (fluorescent proteins/small epitopes) to endogenous proteins in neurons (*Gao et al., 2019*; *Suzuki et al., 2016*); however, such methods are sensitive to INDEL mutations during the KI process. This culminates in reduced KI efficiency and generally restricts the locations where tags can be inserted within the target protein, typically limited to the C-terminal region. Although recent studies have shown that residual HDR activity or introducing exogenous recombinase in neurons is capable of inducing KI (*Cai et al., 2019*; *Nishiyama et al., 2017*), the efficiency of such methods remains reasonably low, prompting us to develop alternative strategies for KI to neurons.

Here, we introduce a CRISPR/Cas9 based method termed TKIT guides to improve the efficiency and reliability of KI to post-mitotic cells and dramatically increasing the flexibility of where tags can

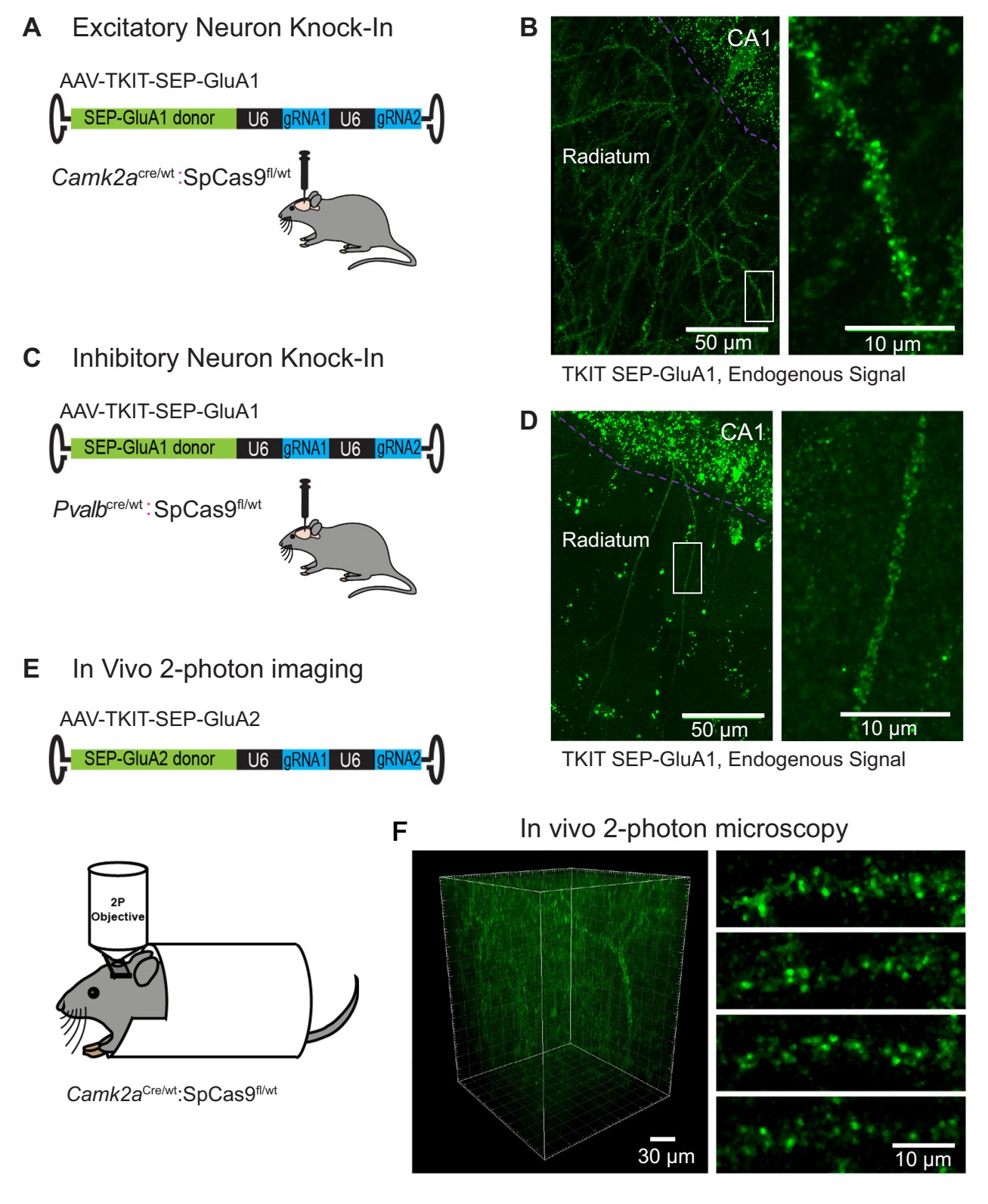

**Figure 6.** Cell-type specific KI with TKIT. (**A**) Schematic of experimental design for targeting excitatory neurons using *Camk2a*[Cre]:Cas9[fl/wt] animals (**B**) Super-resolution Airyscan confocal image of a CA1 pyramidal neuron with TKIT-mediated SEP-GluA1 KI illustrated as a maximum intensity projection. The CA1 cell layer is illustrated with a magenta dotted line with dendrites in Stratum radiatum. A zoomed in region (right) shows the dendritic segment indicated with a white box. (**C**) Schematic of experimental design for targeting PV positive interneurons using *Pvalb*[Cre/wt]:Cas9[fl/wt] animals. (**D**) Super-

*Figure 6 continued on next page*

*Figure 6 continued*

resolution Airyscan confocal image of a CA1 PV neuron with TKIT-mediated SEP-GluA1 KI illustrated as a maximum intensity projection. The CA1 cell layer is illustrated with a magenta dotted line with dendrites in Stratum radiatum. A zoomed in region (right) shows the dendritic segment indicated with a white box. (E) Schematic of experimental setup for two-photon image of virus-mediated TKIT GluA2 KI. (F) In vivo two-photon 3D reconstruction of a layer 2/3 pyramidal neuron in barrel cortex with TKIT-mediated SEP-GluA2 KI. High-magnification images of dendritic segments (right) from a maximum intensity projection of the same cell. Scale bars are indicated in the bottom right corner of images.

be inserted to target proteins. Using TKIT, a tag can be inserted into any user-defined location without undesired amino acid loss or addition. By using a pair of guide RNAs instead of a single guide RNA, TKIT utilizes the use of Cas9 cutting sites that lie in non-coding regions of the gene of interest (such as 5'-UTR and introns). This approach is significantly less sensitive to INDEL mutations since no coding regions of the gene are cut. Even if INDELs are introduced around the cut sites, it is highly likely that the mRNA and subsequent protein are left intact. In addition, because non-coding DNA regions are targeted, and guides do not have to be close to the desired insertion site, there is a greatly increased pool of potential guide RNAs to choose from, enabling high-ranked guides with high on-target score and low off-target probability to be picked. Finally, the switch-and-flip design built into the donor DNA sequence helps to bias the insertion of donor DNA into the correct orientation, further increasing KI efficiency.

We demonstrate that TKIT can be used to tag different AMPA and NMDA receptor subunits, as well as the cytosolic protein gephyrin, with efficiencies of up to 41.7% (GFP-gephyrin). The KI efficiency of SEP-GluA2 reached 16.1% in vivo with a two-virus system. This is greater than the efficiency achieved with the original demonstration of HITI (*Suzuki et al., 2016*; *Nishiyama et al., 2017*). Furthermore, because of the increased protection from INDEL mutations TKIT is particularly useful for tagging proteins at the N-terminus. Exploiting this, we were able to successfully KI SEP to the N-terminus of AMPA and NMDA receptor subunits, positioning the SEP-tag on the extracellular portion of the receptor so the pH sensitivity of SEP (*Miesenböck et al., 1998*) can be utilized to visualize the surface pool of receptors.

While TKIT is resistant to INDELs in the coding region of the gene being edited, the method is not immune to other caveats. For example, if the donor DNA (with the tag/label) is not present to replace the excised genomic DNA then a KO allele could be generated. For GluA2 tagging with SEP we observed KO of GluA2 in 12.4% of transfected cells. Therefore, it will be important to consider that KI neurons are likely surrounded by a small fraction of neurons that are heterozygous or homozygous KO's for the gene of interest.

We have demonstrated that TKIT works in vivo following in utero electroporation, as well as in adult mice following AAV injection. We show that AMPAR subunits can be tagged in wild-type mice with a dual-AAV system, as well as in a cell-type specific manner using transgenic mice. In proof of principle experiments, we showed that TKIT KI of SEP to GluA2 is compatible with two-photon imaging in vivo, enabling us to visualize the surface expression of endogenous GluA2-containing AMPARs in live animals for the first time. This opens up possibilities for tagging and studying endogenous proteins (such as AMPARs) in wild-type or disease model mice (dual virus approach), or in a cell-type-specific manner in Cre-driver mouse lines. While we observed a lower efficiency of KI in vivo (16.1%) compared to in vitro (38.9%) with SEP-GluA2, it is likely that we are under-representing the in vivo efficiency, as a proportion of the cells that were transduced with SpCas9 might have failed to receive the TKIT donor DNA.

Tagging endogenous proteins to study their function and subcellular localization is preferential to exogenous protein overexpression. Therefore, the adoption of endogenous KI strategies such as TKIT will dramatically aid the study of proteins expressed at physiological levels. A potential application of TKIT is its combination with modern proximity labeling techniques such as APEX, BioID, and TurboID (*Branon et al., 2018*; *Rhee et al., 2013*; *Roux et al., 2012*). Those techniques crucially rely on proper subcellular targeting of fusion proteins. Tagging endogenous proteins with proximity labeling enzymes will reduce the confound of nonspecific subcellular protein localization associated with protein overexpression and increase the signal to noise ratio of 'hits' as there will be lower background from mis-localized proteins. However, the feasibility of coupling TKIT with proximity labeling remains to be tested and will likely depend upon the transfection or transduction efficiency, the KI efficiency for the target protein, as well as the absolute abundance of the target protein being

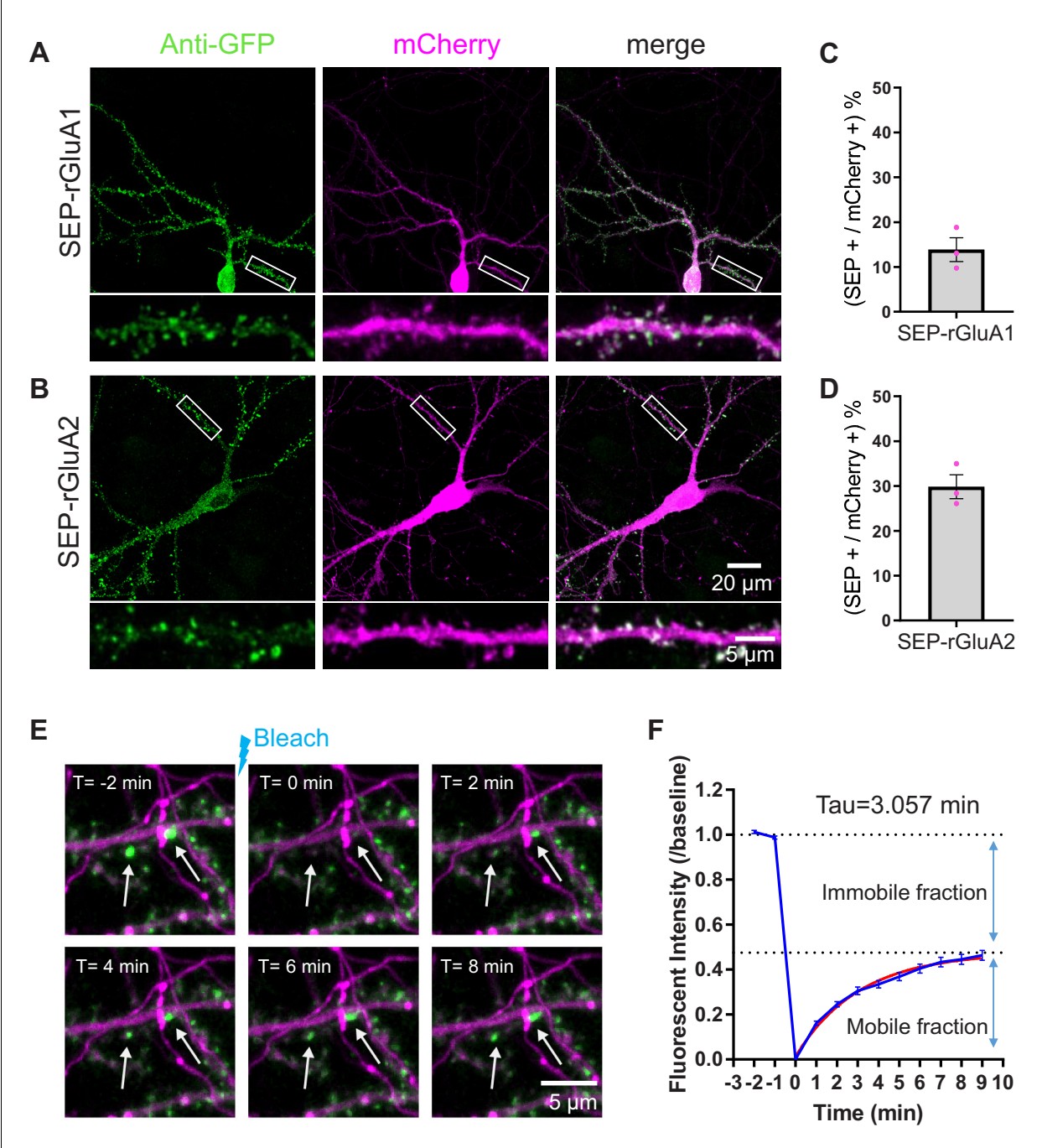

**Figure 7.** TKIT KI and FRAP in rat neurons. TKIT-mediated KI of SEP-GluA1 (**A**) and SEP-GluA2 (**B**) in primary cultured rat neurons. Efficiency of KI for SEP-GluA1 (**C**) and SEP-GluA2 (**D**). (**E**) Representative example of a SEP-GluA1 containing spine undergoing bleaching and fluorescence recovery. The white arrow indicates the spine that will be bleached. Images are separated by 2 min intervals. Scale bar indicates 5 μm. (**F**) Summary of SEP-GluA1 FRAP data with estimated recovery time and immobile fraction of receptors. Red trace is from exponential (one-phase decay) fitting, GraphPad Prism 7. The online version of this article includes the following source data and figure supplement(s) for figure 7:

**Source data 1.** Efficiency counts to assess TKIT knockin of SEP-GluA1 and SEP-GluA2 in rat neurons.
**Source data 2.** Fluorescence recovery after photobleaching data to assess SEP-GluA1 mobility in rat neurons.
**Figure supplement 1.** TKIT of SEP-GluA1 and SEP-GluA2 KI in rat neurons.

tagged. In addition, the flexibility with where tags can be introduced will be of great value as it will allow known protein interaction motifs (i.e., C-terminal PDZ ligands) to be avoided when tagging target proteins, further preventing mis-localization of the protein being studied.

While we detect faithful mRNA splicing of GluA2 following TKIT SEP KI, it remains a potential concern that targeting non-coding regions could have unforeseen effects on mRNA processing. A greater understanding of the biology underlying mRNA processing will facilitate the design of guides to avoid such effects. Given the high flexibility in guide RNA choice, this concern will be easily overcome, as long as genomic DNA cutting sites can be identified that leave mRNA processing unaffected.

In summary, we demonstrate that TKIT is an efficient, reliable, and easy to implement method for tagging endogenous proteins in vitro and in vivo. TKIT can edit mouse and rat genomic DNA, with promised utility in all model organisms. Although, to date, we have only used TKIT to KI tags to proteins, a potential application to clinical therapy to correct causative mutations in genetic disorders could be possible.

# Materials and methods

**Key resources table**

| Reagent type (species) or resource | Designation | Source or reference | Identifiers | Additional information |
|---|---|---|---|---|
| Gene (*Mus musculus*) | GluA1 | GenBank; Ensembl | ID: 14799; ENSMUSG00000020524 | |
| Gene (*Mus musculus*) | GluA2 | GenBank; Ensembl | ID: 14800; ENSMUSG00000033981 | |
| Gene (*Mus musculus*) | GluA3 | GenBank; Ensembl | ID: 53623; ENSMUSG00000001986 | |
| Gene (*Mus musculus*) | GluN1 | GenBank; Ensembl | ID: 14810; ENSMUSG00000026959 | |
| Gene (*Mus musculus*) | GluN2A | GenBank; Ensembl | ID: 14811; ENSMUSG00000059003 | |
| Gene (*Mus musculus*) | GluN2B | GenBank; Ensembl | ID: 14812; ENSMUSG00000030209 | |
| Gene (*Mus musculus*) | Gephyrin | GenBank; Ensembl | ID: 268566; ENSMUSG00000047454 | |
| Gene (*Mus musculus*) | beta2 | GenBank; Ensembl | ID: 14401; ENSMUSG00000007653 | |
| Gene (*Rattus norvegicus*) | rGluA1 | GenBank | ID: 50592 | |
| Gene (*Rattus norvegicus*) | rGluA2 | GenBank | ID: 29627 | |
| Strain, strain background (*Mus musculus*, male and female) | *Camk2a*-Cre mouse | Jackson Laboratory strain 005359 | IMSR Cat# JAX:005359, RRID:IMSR_JAX:005359 | |
| Strain, strain background (*Mus musculus*, male and female) | *Pvalb*-Cre mouse | Jackson Laboratory strain 008069 | IMSR Cat# JAX:008069, RRID:IMSR_JAX:008069 | |
| Strain, strain background (*Mus musculus*, male and female) | Cas9fl/fl mouse | Jackson Laboratory strain 027632 | IMSR Cat# JAX:027632, RRID:IMSR_JAX:027632 | |
| Genetic reagent (*Streptococcus pyogenes*) | px330 | RRID:Addgene_42230 | RRID:Addgene_42230 | |
| Genetic reagent (*Streptococcus pyogenes*) | pAAV-nEF-SpCas9 | RRID:addgene_87115 | RRID:Addgene_87115 | |
| Genetic reagent (*Escherichia coli*) | pMiniT Vector | NEB | E1202 | NEB replaced this vector with pMiniT 2.0 Vector |
| Genetic reagent (*Mus musculus*) | AAV2/9-TKIT-SEP-GluA1 | Viral Core at the Janelia Research Campus | | AAV for in vivo delivery TKITdonor and two guides |

*Continued on next page*

*Continued*

| Reagent type (species) or resource | Designation | Source or reference | Identifiers | Additional information |
|---|---|---|---|---|
| Genetic reagent (*Mus musculus*) | AAV2/9-TKIT-SEP-GluA2 | Viral Core at the Janelia Research Campus | | AAV for in vivo delivery TKITdonor and two guides |
| Genetic reagent (*Streptococcus pyogenes*) | AAV2/9-nEF-HA-spCas9 | Viral Core at the Janelia Research Campus | | AAV for in vivo delivery TKITdonor and two guides |
| Genetic reagent (*Aequorea victoria*) | AAV2/9.CMV.PI.EGFP. WPRE.bGH | RRID:addgene_105530 | RRID:Addgene_105530 | |
| Biological sample (*M. musculus*) | Primary neuron culture | | | isolated from E18 mouse embryo and cultured |
| Biological sample (*R. norvegicus*) | Primary neuron culture | | | isolated from E18 rat embryo and cultured |
| Antibody | Anti-GluA1 (Rabbit polyclonal) | Homemade | JH4294 | IF (1:200) |
| Antibody | Anti-GluA2/3 (Rabbit polyclonal) | Homemade | JH4854 | IF (1:250) |
| Antibody | Anti-GluA2 (mouse monoclonal) | Eric Gouaux Lab | clone 15F1 | IF (1:5000) |
| Antibody | Anti-GluN1 (mouse monoclonal) | NeuroMab, clone N308/48 | UC Davis/NIH NeuroMab Facility Cat# N308/48, RRID:AB_2877408 | IF (1:200) |
| Antibody | Anti-GluN2a (Rabbit polyclonal) | Homemade | JH6097 | IF (1:200) |
| Antibody | Anti-GluN2b (Rabbit polyclonal) | Millipore, AB1548 | Millipore Cat# AB1548, RRID:AB_90759 | IF (1:100) |
| Antibody | Anti-beta2 (mouse monoclonal) | Millipore, MAB341 | Millipore Cat# MAB341, RRID:AB_2109419 | IF (1:200) |
| Antibody | Anti-PSD95 (mouse monoclonal) | NeuroMab, clone K28/43 | UC Davis/NIH NeuroMab Facility Cat# K28/43, RRID:AB_2877189 | IF (1:2500) |
| Antibody | Anti-Gephyrin (mouse monoclonal) | Synaptic systems, cat# 147011 | Synaptic Systems Cat# 147 011, RRID:AB_887717 | IF (1:2500) |
| Antibody | Anti-GFP (chicken polyclonal) | Abcam, cat # ab13970 | (Abcam Cat# ab13970, RRID:AB_300798) | IHC (1:5000) |
| Antibody | Anti-GFP (chicken polyclonal) | AVES, cat# GFP-1020 | Aves Labs Cat# GFP-1020, RRID:AB_10000240 | IF (1:5000) |
| Antibody | Anti-myc (mouse monoclonal) | Homemade | Clone 9E10 | IF (1:1000) |
| Antibody | Anti-HA (Rabbit monoclonal) | Cell Signaling, clone C29F4 | Cell Signaling Technology Cat# 3724, RRID:AB_1549585 | IF (1:500) |
| Recombinant DNA reagent (*Escherichia coli*) | pSG | This paper | | For the guide2 cloning |
| Recombinant DNA reagent (*Streptococcus pyogenes*) | GluA1 (px330:2-guide-Cas9) | This paper | | Twoguides for GluA1 (mouse) in px330 |
| Recombinant DNA reagent (*Mus musculus*) | pMiniT:SEP-GluA1 TKIT donor | This paper | | TKIT donor for SEP-GluA1 (mouse) in pMiniT |
| Recombinant DNA reagent (*Mus musculus*) | pMiniT:myc-GluA1 TKIT donor | This paper | | TKIT donor for myc-GluA1 (mouse) in pMiniT |
| Recombinant DNA reagent (*Streptococcus pyogenes*) | GluA2 (px330:2-guide-Cas9) | This paper | | Two guides for GluA2 (mouse) in px330 |
| Recombinant DNA reagent (*Mus musculus*) | pMiniT:SEP-GluA2 TKIT donor | This paper | | TKIT donor for SEP-GluA2 (mouse) in pMiniT |
| Recombinant DNA reagent (*Mus musculus*) | pMiniT:tdTomato-GluA2 TKIT donor | This paper | | TKIT donor for tdTomato-GluA2 (mouse) in pMiniT |

*Continued on next page*

*Continued*

| Reagent type (species) or resource | Designation | Source or reference | Identifiers | Additional information |
|---|---|---|---|---|
| Recombinant DNA reagent (*Mus musculus*) | pMiniT:myc-GluA2 TKIT donor | This paper | | TKIT donor for myc-GluA2 (mouse) in pMiniT |
| Recombinant DNA reagent (*Streptococcus pyogenes*) | GluA3 (px330:2-guide-Cas9) | This paper | | Two guides for GluA3 (mouse) in px330 |
| Recombinant DNA reagent (*Mus musculus*) | pMiniT:SEP-GluA3 TKIT donor | This paper | | TKIT donor for SEP-GluA3 (mouse) in pMiniT |
| Recombinant DNA reagent (*Streptococcus pyogenes*) | GluN1 (px330:2-guide-Cas9) | This paper | | Two guides for GluN1 (mouse) in px330 |
| Recombinant DNA reagent (*Mus musculus*) | pMiniT:SEP-GluN1 TKIT donor | This paper | | TKIT donor for SEP-GluN1 (mouse) in pMiniT |
| Recombinant DNA reagent (*Streptococcus pyogenes*) | GluN2A (px330:2-guide-Cas9) | This paper | | Two guides for GluN2A (mouse) in px330 |
| Recombinant DNA reagent (*Mus musculus*) | pMiniT:SEP-GluN2A TKIT donor | This paper | | TKIT donor for SEP-GluN2A (mouse) in pMiniT |
| Recombinant DNA reagent (*Streptococcus pyogenes*) | GluN2B (px330:2-guide-Cas9) | This paper | | Two guides for GluN2B (mouse) in px330 |
| Recombinant DNA reagent (*Mus musculus*) | pMiniT:SEP-GluN2B TKIT donor | This paper | | TKIT donor for SEP-GluN2B (mouse) in pMiniT |
| Recombinant DNA reagent (*Streptococcus pyogenes*) | Gephyrin (px330:2-guide-Cas9) | This paper | | Two guides for Gephyrin (mouse) in px330 |
| Recombinant DNA reagent (*Mus musculus*) | pMiniT:GFP-Gephyrin TKIT donor | This paper | | TKIT donor for GFP-Gephyrin (mouse) in pMiniT |
| Recombinant DNA reagent (*Mus musculus*) | pMiniT:tdTomato-Gephyrin TKIT donor | This paper | | TKIT donor for tdTomato-Gephyrin (mouse) in pMiniT |
| Recombinant DNA reagent (*Streptococcus pyogenes*) | beta2 (px330:2-guide-Cas9) | This paper | | Two guides for beta2 (mouse) in px330 |
| Recombinant DNA reagent (*Mus musculus*) | pMiniT:SEP-beta2 TKIT donor | This paper | | TKIT donor for SEP-beta2 (mouse) in pMiniT |
| Recombinant DNA reagent (*Streptococcus pyogenes*) | rGluA1 (px330:2-guide-Cas9) | This paper | | Two guides for rGluA1 (mouse) in px330 |
| Recombinant DNA reagent (*Rattus norvegicus*) | pMiniT:SEP-rGluA1 TKIT donor | This paper | | TKIT donor for SEP-rGluA1 (mouse) in pMiniT |
| Recombinant DNA reagent (*Streptococcus pyogenes*) | rGluA2 (px330:2-guide-Cas9) | This paper | | Two guides for rGluA2 (mouse) in px330 |
| Recombinant DNA reagent (*Rattus norvegicus*) | pMiniT:SEP-rGluA2 TKIT donor | This paper | | TKIT donor for SEP-rGluA2 (mouse) in pMiniT |
| Recombinant DNA reagent (*Mus musculus*) | pAAV-SEP-GluA1 TKIT | This paper | | TKIT donor and two guides for SEP-GluA1 (mouse) in one pAAV vector |
| Recombinant DNA reagent (*Mus musculus*) | pAAV-SEP-GluA2 TKIT | This paper | | TKIT donor and two guides for SEP-GluA2 (mouse) in one pAAV vector |
| Commercial assay or kit | NEBuilder HiFi DNA Assembly Cloning Kit | NEB | cat#: E5520 | |
| Software, algorithm | Fiji/ImageJ | Fiji | Fiji, RRID:SCR_002285 | |
| Software, algorithm | GraphPad Prism 7.0 | GraphPad Software | GraphPad Prism, RRID:SCR_002798 | |
| Software, algorithm | Benchling CRISPR tool | http://www.benchling.com | Benchling, RRID:SCR_013955 | |
| Software, algorithm | IMARIS 9.6.0 (Bitplane) | BitPlane | Imaris, RRID:SCR_007370 | |
| Other | Prolong Glass Antifade mounting media | Thermo Scientific | P36984 | |

## Animals

All animals were treated in accordance with the Johns Hopkins University Animal Care and Use Committee guidelines. For all experiments, animals were kept on a 12 hr:12 hr light/dark cycle. In utero electroporation and viral injection experiments were performed using male and female C57BL/6N mice (Charles River). For cell-type-specific viral injection experiments we crossed transgenic *Camk2a*-Cre (T29-1, Jackson Laboratory, strain 005359) or *Pvalb*Cre (Jackson Laboratory, strain 008069) mice with Cas9fl/fl animals (Jackson Laboratory, strain 027632). Sprague Dawley rats were purchased from Envigo.

## Culture of mouse/rat primary neurons

Primary neurons were obtained from the cortex of E18 C57BL/6 mouse or Sprague Dawley rat brains. All dissection procedures were performed in ice cold dissection medium. Cortical tissue was dissociated with papain digestion and gentle trituration, and 300 k cells/well (12-well plates) were plated on poly-L-lysine pre-coated coverslips (18 mm) with Neurobasal plus media (Gibco) supplemented with 2% B27 (Gibco), 1% Glutamax (Gibco), 1% penicillin/streptomycin, and 5% horse serum. The cultures were incubated at standard conditions (37°C in humidified 5% CO2/95% air atmosphere). The cultures were fed twice per week with Neurobasal media (Gibco) with 2% B27, 1% Glutamax (Gibco), and 1% penicillin/streptomycin.

## DNA constructs/plasmids

All guides were designed using the Benchling CRISPR tool (http://www.benchling.com). To construct dual-promotor guide RNA expression vectors, a pair of complementary 20 bp oligos for each guide RNA sequence was annealed and cloned into pX330 (Addgene 42230; for guide 1) or pSG (a shuttle vector; for guide 2) separately. Both constructs were confirmed by sequencing. To generate one vector with the two guides, the pSG-guide2 vector was digested with NheI/KpnI and the ~430 bp fragment (U6 promotor and single guide RNA expression cassette for guide 2) was gel purified and ligated to pX330-guide1 at XbaI/KpnI sites. All CRISPR/Cas9 guide RNA sequences and PAM sequences used in this study are shown in *Supplementary file 1*.

To construct each TKIT-donor plasmid, three DNA fragments were generated with PCR from genomic DNA or plasmids using Q5 High-Fidelity DNA Polymerase (NEB Cat# M0491). Fragment 1 included the genomic DNA between guide 1 and the KI site plus flipped guide 2 (with PAM) at its 5'-end; fragment 2 included the fluorescent protein/myc tag with linkers on both sides; and fragment 3 included the genomic DNA between the KI site and guide 2 plus flipped guide 1 (with PAM) at its 3'-end. These three DNA fragments were assembled into a vector backbone (NEB pMiniT vector, EcoRI linearized) with the NEBuilder HiFi DNA Assembly Cloning Kit (NEB Cat# E5520). All TKIT donors' sequence were confirmed with pMiniT forward or reverse sequencing primers. All PCR primers used for constructing TKIT donors were shown in *Supplementary file 2*.

While we used the above approach to generate all plasmids used in this study, we have since developed an expedited method for Cas9-2-guide cloning. In this method, we use a four-fragment HiFi assembly. Fragment 1 (vector) is made by digesting pX330 with PciI and Acc65I. Fragment 2, made by PCR, is generated using the common forward primer 5'-tggccttttgctggccttttgctcacatgt-GAGGGCCTATTTCCCATG-3' and the custom reverse primer 5'-tagctc-taaaacNNNNNNNNNNNNNNNNNNNNCGGTGTTTCGTCCTTTCCAC-3'. The red region indicates the guide 1 antisense strand ending with 'C '(total # of N's should be 19 if guide 1 starts with 'G' or 20 if guide 1 does not start with G). This product is made using pX20 as a template. Fragment 3, made by PCR, using a design-specific forward primer.

5'-aggacgaaacaccGNNNNNNNNNNNNNNNNNNNNGTTTTAGAGCTAGAAATAGCAAGTTaaaa-taag-3' where the red region is the guide 1 sense strand beginning with 'G' (total # of N's should be 19 if guide 1 starts with G or 20 if guide 1 does not start with G) and the design-specific reverse primer.

5'-tagctctaaaacNNNNNNNNNNNNNNNNNNNNCGGTGTTTCGTCCTTTCCAC-3' where the red region is guide 2 antisense strand ending with 'C '(total # of N's should be 19 if guide 1 starts with 'G' or 20 if guide 1 does not start with G). This product is made using pX52 (modified from pX330) as a template. Fragment 4, made by PCR, using the design-specific forward primer.

5'-aggacgaaacaccGNNNNNNNNNNNNNNNNNNNNGTTTTAGAGCTAGAAATAGCAAG-3' where red is the Guide 2 sense strand beginning with 'G' (total # of N's should be 19 if guide 1 starts with G or 20 if guide 1 does not start with G) and the common reverse primer 5'-ccatttaccgtaagttatg-taacgggtaccgatatctagaaaaaagcaccg-3'. This product is made using pX20 (modified from pX330) as a template. These four fragments can then be assembled, and bacteria transformed with the resulting product.

Furthermore, as an additional alternative, it would also be possible to instead utilize pX333 (Addgene plasmid #64073), which enables the cloning of two guides with two unique restriction sites.

All constructs described in this manuscript will be available to academic researchers through Addgene.

## Reverse transcription from primary neuron culture

To analyze knock-in cDNA, neurons from a single well (6-well plate) of electroporated neuronal culture were lysed at DIV19 and RNA was isolated using the RNAqueous-Mico Total RNA Isolation kit (Invitrogen Cat # AM1931). This RNA was then converted to cDNA using the Superscript III First Strand Synthesis System (Invitrogen Cat #18080051). PCR followed by Sanger sequencing analysis was then performed to confirm expression of correctly inserted SEP into GluA1 and GluA2 mRNA. For SEP-GluA1 the upstream junction between GluA1 and pHluorin was amplified using a sense primer in the GluA1 5' UTR (5-TTTCTGCACCGGTTTTCTAGGT-3') and an antisense primer located within SEP (5'-AGTAGTGACAAGTGTTGGCCAAG-3'). The downstream junction was amplified using a sense primer in SEP (5'-GGCGTTCAACTAGCAGACCATTA-3') and an antisense primer (5'-ATGTG TCAACGGGAAAACTTGGA-3') located in exon 2 of GluA1. Sequence analysis of these fragments demonstrated that SEP-GluA1 mRNA is being expressed and there is proper splicing of RNA between exon 1, containing the SEP insertion, and exon 2 of GluA1. For SEP-GluA2 the upstream junction between GluA2 and SEP was amplified using a sense primer in the GluA2 5' UTR (5'-ACGAGGGGATATTTTGTGGATGC-3') and the antisense SEP primer described above. The downstream junction was amplified using the sense primer in SEP described above and an antisense primer (5'-CTGAACCATCCCTACCCGAAATG-3') located in exon 2 of GluA2. Sequence analysis of these fragments also confirmed that SEP-GluA2 is being expressed and the RNA is spliced correctly.

## Transfection of in vitro cultured cells

Lipofectamine 2000 (Invitrogen) was used for transfection of mouse and rat primary neurons. Transfection complexes were prepared following the manufacturer's instructions.

## Immunocytochemistry of primary neurons

Cells were fixed in 4% paraformaldehyde (PFA) with 4% sucrose at room temperature for 15 min. Then cells were permeabilized with 0.25% Triton X-100 in PBS for 10 min and blocked with 5% bovine serum albumin (BSA) in PBS for 60 min with slow shaking at room temperature. Primary antibodies were diluted in 2.5% BSA/PBS and cells were incubated overnight at 4°C in a humidified chamber with primary antibodies anti-GFP (Abcam or Aves, 1:5000), anti-GluA1 (homemade, 1:200), anti-GluA2/3 (homemade, 1:250), anti-NR1 (homemade, 1:200), anti-NR2A (homemade, 1:200), anti-NR2B (Millipore, 1:100), anti-PSD95 (Neuromab, 1:2500), anti-Gephyrin (Synaptic Systems, 1:2500), anti-myc (homemade, 1:1000), and anti-beta2 (Millipore, 1:200). The next day, cells were washed with PBS three times, and incubated for 1 hr at room temperature with corresponding secondary antibodies Alexa Fluor 405, 488, 568, and 647 (Thermo Fisher, dilutions 1:500–1000). After a second round of washing with PBS, the cells were mounted using mounting media (PermaFluor, Thermo Scientific) and allowed to dry overnight at room temperature. Coverslips were imaging immediately or stored at 4°C for subsequent imaging.

## Image capture and measurement of knock-in efficiency

Live-cell imaging was performed with mouse neuronal culture at DIV16 with a LSM 880 confocal microscope (Carl Zeiss). Coverslips containing neurons were assembled in a custom-made chamber filled with artificial cerebrospinal fluid (ACSF; 25 mM HEPES, 120 mM NaCl, 5 mM KCl, 2 mM CaCl$_2$, 2 mM MgCl$_2$, pH 7.4). The SEP fluorescence was imaged at 488 nm excitation and collected through

493–553 nm, while the mCherry signal was imaged at 561 nm excitation and collected through 559–654 nm emission. Neurons were imaged through a 63× oil objective (N.A. = 1.40). Images were analyzed using Fiji/ImageJ (*Schindelin et al., 2012*).

After immunostaining with GFP antibody (Abcam cat # ab13970, 1:5000) for SEP/GFP KI, coverslips were imaged with a Zeiss LSM 880 microscope using Zen Black software and identical exposure parameters within each knock-in group. At least three coverslips were used per experimental group. Z-stacks of mCherry positive cells were acquired (1024 × 1024 pixels, with at least nine optical slices per z-stack with interval of 0.5 µm). Images were analyzed later in Fiji/ImageJ. To calculate the efficiency of TKIT-mediated targeted gene KI, number of GFP positive cells was divided by number of mCherry positive cells. To calculate the KI/WT/KO fraction, we analyzed mCherry positive cells in TKIT SEP-GluA2 transfected neurons in Fiji. For GFP and mCherry both positive cells, we considered those as successful KI cells. For the rest of mCherry positive cells, ROIs were drawn on the soma of cells based on mCherry signal, maxiROIs (cover the whole image) were also drawn on the same images. Fluorescent intensity of anti-GluA2 antibody was quantified in Fiji after subtracting background. The fluorescent intensity of maxiROI was set as a threshold to divide WT and KO cells based on their anti-GluA2 signal. To detect GluA2 expression in mouse cortical neurons, without TKIT manipulation, we fixed DIV 15 neurons grown on glass coverslips (as described above). These cells had been transduced with AAV2/9.CMV.PI.EGFP.WPRE.bGH (Addgene ID 105530) at DIV2 to serve as a morphology marker for neuron identification. Using IMARIS 9.6.0 (Bitplane), a surface was rendered at the cell body of GFP cells. Surfaces were excluded if the morphology of the cell was not consistent with a neuron. The surface volume was then used as a mask to calculate the average signal of GluA2, identified with immunofluorescence (as described above but with an N-terminal GluA2 antibody, clone 15F1).

## Quantification of colocalization

To quantify the degree of colocalization between SEP-GluAs and anti-GluAs (GluA1, GluA2, and GluA3) fluorescent signal, all dendritic spines on a 50–100 µm dendrite from each knock-In neuron were analyzed in Fiji. To avoid bias, ROIs were draw based on mCherry signal, without seeing SEP-GluAs and anti-GluAs signal intensity. Fluorescent intensity was quantified in Fiji after subtracting background. Correlation (slope) between SEP and anti-GluAs fluorescent intensity was calculated by linear fitting (GraphPad Prism 7). To access the degree of colocalization between SEP-GluAs and anti-PSD95, the same correlation analysis with above was done between SEP and anti-PSD95 fluorescent intensity.

## AAV design and production

To construct AAV vectors, both dual-U6 promoter expression cassette and donor were amplified from pX330-TKIT plasmid and corresponding TKIT donor plasmid, separately by PCR primers (*Supplementary file 2*). Then these two fragments were assembled into pAAV backbones with the NEBuilder HiFi DNA Assembly Cloning Kit. Resulting clones were confirmed by sequencing. Plasmid pAAV-nEF-SpCas9 is from Addgene (#87115) (*Suzuki et al., 2016*). All AAVs were packaged with serotypes 2/9 and were generated by the Viral Core at the Janelia Research Campus.

## In utero electroporation

Cortical layer 2/3 neurons in the mouse barrel cortex were transfected with TKIT-SEP-GluA2-donor and GluA2-guides-SpCas9 in a 2:1 ratio by targeted in utero electroporation in E15 embryos as previously described (*Saito and Nakatsuji, 2001*; *Zhang et al., 2015*). Following anesthesia with Avertin and laparotomy, the DNA plasmid mixture (0.5–1 µl) was injected into the lateral ventricle of embryos using a pulled-glass pipette. Electroporation was performed using bipolar tweezer electrodes (CUY650P5, Nepagene, Japan) and a square pulse generator (CUY21, BEX Co, LTD., Japan). Previously established electrode positioning was used to target the barrel cortex.

## Virus injection surgery

Animals were anaesthetized with Isoflurane (Kent Instruments), secured in a stereotaxic frame (Kopf), and their body temperature monitored and maintained at 37°C with a closed-loop temperature control system (Kent Instruments). The skin above the skull was cleaned with ethanol wipes and the hair

removed. Animals were injected with sterile saline (VetOne; 0.5 ml), buprenorphine (ZooPharm; 1 mg/kg), and lidocaine (VetOne; 2%) all subcutaneous, with lidocaine injected under the skin over the skull. An incision was made to expose the skull, and a craniotomy performed to expose the brain surface. Glass pipettes (Drummond Science Company; Wiretrol II) were pulled (Sutter Instruments) and sharpened to a 30° angle (Medical Systems Corp) and used for controlled virus injection. A pneumatic injector (Narishige) was used to inject viruses at a rate of 100 nl/min. After each injection the pipette was kept in place for 5 min before being raised to the next injection depth, or being removed slowly from the brain. To target the dorsal hippocampus the following stereotaxic coordinates and injection volumes were followed, relative to bregma and pia. AP: −2, ML: −1.5, Z: 1.5 (200 nl), −1.3 (200 nl) and −1 (300 nl) and AP: −2.1, ML: 2.5, Z: 2.2 (300), −2 (300),–1.75 (200). Viruses, both packaged by the Janelia Research Campus Viral Core, were not diluted before injection and had the following titers: AAV2/9-TKIT-SEP-GluA1 ($1.21E^{14}$ GC/ml) and AAV2/9-TKIT-SEP-GluA2 ($9E^{13}$ GC/ml).

For the dual virus experiments AAV2/9-TKIT-SEP-GluA2 was mixed 1:1 with AAV2/9-nEF-HA-spCas9 ($1.121E^{13}$ GC/ml) and injected into the right hemisphere of C57BL/6N mice aged 6 weeks. The injection coordinates were the same as above but the injection volume was increased to 400 nl per site. After injection the skin was sutured (Ethicon) and sealed with glue (VetBond). Animals were placed in a heated cage to recover with access to softened food.

## Window surgery

In adult mice, a custom cut square 3 × 3 mm cranial window (#1 coverslip glass) was placed over the cortex of previously electroporated or virally injected mice as described previously (*Roth et al., 2020*). Mice were anesthetized using Avertin and the anti-inflammatory drugs Carprofen and Dexamethasone were administered. A craniotomy matching the size of the coverslip was cut using #11 scalpel blades (Fine Science Tools) and the coverslip was carefully placed on top of the dura within the craniotomy without excessive compression of the brain. The window was centered using stereotactic coordinates 3 mm lateral and 1.5 mm posterior from bregma for barrel cortex. The window and skull were sealed using dental cement (C and B Metabond, Parkell). A custom-made metal head bar was attached to the skull during surgery to fixate the mouse for imaging. Mice were allowed to recover for 2–3 weeks after surgery before two-photon imaging.

## Histology and immunohistochemistry

Mice were deeply anaesthetized and transcardially perfused with PBS followed by 4% PFA (Electron Microscopy Sciences), each ice cold. Brains were post fixed in 4% PFA for exactly 2 hr at 4°C. After washing with PBS, brains were sliced into 60 mm sections on a vibratome (VT-1000, Leica). Sections were either mounted immediately for super-resolution (Airyscan, Zeiss) confocal microscopy (see below), or processed for immunohistochemistry. To amplify the GFP signal in TKIT-SEP-GluA1 KI mice, sections were washed three times in PBS, permeabilized for 20 min in PBST (0.3% Triton-X), and blocked for 1 hr in 5% normal goat serum (NGS) in PBST (0.15% Triton-X) all at room temperature. A chicken anti-GFP antibody (Abcam, Chicken, 1:5000) in 5% NGS and PBST (0.15% Triton-X) was added to the slices and incubated, rocking overnight at 4°C. The following day, slices were washed four times with PBS at room temperature, and then incubated with an anti-chicken Alexa-488 secondary antibody (1:500 titer, Molecular Probes) overnight at 4°C. Finally, sections were washed with PBS four times at room temperature. All slices were mounted Prolong Glass Antifade mounting media (Thermo Scientific). For detection of AAV-mediated spCas9 expression the same immunohistochemistry protocol was followed but with the addition of an antibody against HA (Cell Signaling, Rabbit C29F4, 1:500).

## Confocal imaging in brain slices

All confocal images were taken on a Zeiss LSM 880 microscope using Zen Black software. For Airyscan imaging of endogenous SEP-GluA1 we used the super-resolution (SR) mode and optimal pixel resolution in X, Y, and Z. For estimation of KI efficiency using the dual-virus approach, SEP positive and HA-Cas9 positive cells were counted using the cell counter plugin in Fiji. Counts were made from maximum intensity projections images taken from the CA1 cell layer of the dorsal hippocampus.

## In vivo two-photon imaging

In vivo images were acquired of mice under Isoflurane anesthesia with ZEISS LSM880 two-photon laser-scanning microscope. Layer 2/3 pyramidal neurons were imaged using a 20×/1.0 NA water-immersion objective lens (Zeiss). SEP-GluA2 was excited at 910 nm with a Ti:sapphire laser (Coherent) with ~100 mW of power delivered to the back-aperture of the objective. Image stacks were acquired at 1024 × 1024 pixels with a voxel size of 0.21 μm in x and y and a z-step of 1 μm at a pixel dwell time of ~2 ns. We simultaneously acquired images in the green (495–550 nm emission) and red (575–610 nm emission) channel under 910 nm excitation. The red channel, which shows no signal from SEP fluorophores, was acquired to reduce nonspecific fluorescence signal from in vivo brain tissue. Subtracting red images from green images resulted in images corrected for nonspecific autofluorescence used for representative images. Images shown in figures were median filtered and contrast enhanced. 3D rendered images were created using Imaris (Bitplane).

## Fluorescence recovery after photobleaching

Rat primary cultured hippocampal neurons were transfected at DIV7-8 with TKIT-SEP-GluA1 constructs with mCherry or Azurite as a cell fill. At DIV 21–23 cells were live imaged in ACSF (as above) using a Zeiss LSM 880 microscope using Zen Black software. A time course experiment was conducted with images acquired as z-stacks every 60 s. After two baseline images were acquired a subset of SEP puncta were photobleached (40 iteration at 100% laser power) before a further 10 images were acquired, each 60 s apart. FRAP data was analyzed in Fiji (*Schindelin et al., 2012*). Initially maximum intensity projections were generated from the z-stacks, and the data was median filtered (1 pixel) and registered (rigid body) using the Multi-Stack Reg plugin. From each image two background regions were selected, averaged, and then subtracted from each data point. Data were normalized and plotted in GraphPad Prism for calculation of the recovery kinetics and for estimation of the immobile fraction.

## Acknowledgements

We thank all members of the RLH laboratory for helpful comments, discussion, and critical reading of the manuscript. We are grateful to Shaowen Ju and Orlando Martinez for help with cloning. We thank Adeline JH Yong for help with mouse neuron culture. We also thank Michele Pucak for help with in vivo two-photon imaging. This work was supported by NIH grants RO1 NS036715 and RF1 MH123212 (to RLH) and grant P30 NS050274 to the Johns Hopkins Multiphoton Imaging Core as well as CAMS Innovation Fund for Medical Sciences 2019-I2M-5-054 (to HF)

## Additional information

### Funding

| Funder | Grant reference number | Author |
| --- | --- | --- |
| National Institute of Neurological Disorders and Stroke | RO1 NS036715 | Richard L Huganir |
| National Institute of Mental Health | RF1 MH123212 | Richard L Huganir |
| Chinese Academy of Medical Sciences | CAMS Innovation Fund for Medical Sciences 2019-I2M-5-054 | Huaqiang Fang |

The funders had no role in study design, data collection and interpretation, or the decision to submit the work for publication.

### Author contributions

Huaqiang Fang, Conceptualization, Formal analysis, Validation, Investigation, Visualization, Methodology, Writing - original draft, Writing - review and editing; Alexei M Bygrave, Formal analysis, Validation, Investigation, Visualization, Methodology, Writing - original draft, Writing - review and editing; Richard H Roth, Validation, Investigation, Visualization, Methodology, Writing - review and

editing; Richard C Johnson, Validation, Methodology, Writing - review and editing; Richard L Huganir, Conceptualization, Resources, Supervision, Funding acquisition, Project administration, Writing - review and editing

**Author ORCIDs**
Huaqiang Fang (iD) https://orcid.org/0000-0002-2375-859X
Alexei M Bygrave (iD) https://orcid.org/0000-0003-2291-923X
Richard H Roth (iD) https://orcid.org/0000-0002-6855-999X
Richard L Huganir (iD) https://orcid.org/0000-0001-9783-5183

**Ethics**
Animal experimentation: All of the animals were handled according to approved animal care and use committee (ACUC) protocol (Protocol Number: MO20M336, MO18M36, RA20M14) of Johns Hopkins University.

**Decision letter and Author response**
Decision letter https://doi.org/10.7554/eLife.65202.sa1
Author response https://doi.org/10.7554/eLife.65202.sa2

## Additional files

**Supplementary files**
• Supplementary file 1. Location and sequences of TKIT guides.
• Supplementary file 2. PCR primers for TKIT donors or AAV constructs assembly.

**Data availability**
Source data files have been provided for all figures. Due to the size of the raw image data these will be made available upon request.

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
