## [Decision Letter]

**Acceptance summary:**

This manuscript describes a novel CRISPR-based gene editing strategy that enables molecular tagging and tracking of synaptic proteins in neurons. The results are compelling and demonstrate the ability to tag several different proteins of interest in rat and mouse neurons, as well as in vivo efficacy in the adult rodent brain. This methodology will help to unravel molecular mechanisms critical for synaptic plasticity, organization, and structure, but could easily be modified for imaging of other cellular proteins important for neuronal function.

**Decision letter after peer review:**

Thank you for submitting your article "An Optimized CRISPR/Cas9 Approach for Precise Genome Editing in Neurons" for consideration by *eLife*. Your article has been reviewed by two peer reviewers, one of whom is a member of our Board of Reviewing Editors, and the evaluation has been overseen by Lu Chen as the Senior Editor. The following individual involved in review of your submission has agreed to reveal their identity: Daniel Choquet (Reviewer #3).

The reviewers have discussed the reviews with one another and the Reviewing Editor has drafted this decision to help you prepare a revised submission.

We would like to draw your attention to changes in our policy on revisions we have made in response to COVID-19 (https://elifesciences.org/articles/57162). Specifically, when editors judge that a submitted work as a whole belongs in *eLife* but that some conclusions require a modest amount of additional new data, as they do with your paper, we are asking that the manuscript be revised to either limit claims to those supported by data in hand, or to explicitly state that the relevant conclusions require additional supporting data.

Summary:

This manuscript describes a novel CRISPR/Cas9 approach for endogenous knock-in, which the authors term "Target Knock In with Two guides", or TKIT. The advances of this approach are twofold: (1) the purposeful targeting of non-coding regions for CRISPR sgRNA placement, and (2) a clever CRISPR donor template design which biases towards forward insertion of the donor sequence. As a result, this approach circumvents the notorious inefficiency of non-homologous end joining pathway for desired DNA insertion. The results are compelling and demonstrate the ability to tag several different proteins in vitro in both rat and mouse neurons, as well as in vivo efficacy in proof-of-principle experiments. This methodology will be of interest to researchers interested in synaptic proteins, but could easily be modified for imaging of other cellular proteins that are important for neuronal function.

Essential revisions:

1) The finding that a small fraction of neurons undergo GluA2 KO is puzzling, since no KO alleles were detected in genomic PCR (Figure 4—figure supplement 2). Is it more likely that the same fraction of neurons would lack GluA2 expression in non-transfected controls? Or controls that only receive mCherry plasmid transfections? Complete KO would require successful CRISPR targeting at both alleles, without donor insertion at either one. Either way, if complete KO in some cells is a possibility, this could be investigated more systematically (or at least discussed more thoroughly) since it may influence potential results.

2) Recent reports suggest that AAV vector genomes can integrate at DSBs created by Cas9 (PMIDs: 33087932, 31570731), and Cas9 can often create an opportunity for unintended insertions. Although PCR of genomic DNA (shown in Figure 4—figure supplement 2) detected both WT and KI alleles, it is not clear how often insertion of other sequences (with variable length) occurs, and no data at the level of DNA sequence is provided following AAV delivery methods. The fidelity of clean donor insertion could be examined by Sanger sequencing of genomic DNA clones to more completely define the results of insertion and genomic sequence changes around cut sites. This would also help to identify potential issues that may affect non-coding sequences (which, at the locations targeted here, are likely to be important regulatory elements).

3) I agree with the authors that tagging endogenous proteins is likely to preserve the normal physiological processes of transcription, mRNA processing, and protein localization, all of which will be specific to the target in question. Thus, the ability to use this system with proximity labeling proteomics approaches is a promising future direction. However, the author's claims about the superiority of this combination in the Discussion section should likely be tempered given that the tags used by these approaches were not incorporated or tested here.

4) Was the sequential sgRNA cloning and ligation process used for all genes targeted here? While it clearly works, this is somewhat laborious and may discourage others from adopting this technology. Since all TKIT targets will require two sgRNAs, the authors might consider using pX333 (PMID: 25337876; Addgene plasmid 64073) instead of pX330. This would enable more rapid sgRNA insertion (due to two unique restriction digest sites) instead of sequential amplification, gel purification, and ligation.

5) Regarding the FRAP experiments, could the authors expand their discussion on the difference between their measurements obtained on endogenous levels of receptors and those previously obtained on overexpressed receptors.

6) In the cell specific approach, could the author provide feedback on Cas9 cell expression versus viral Cas9 expression.

---

## [Author Response]

Essential revisions:1) The finding that a small fraction of neurons undergo GluA2 KO is puzzling, since no KO alleles were detected in genomic PCR (Figure 4—figure supplement 2). Is it more likely that the same fraction of neurons would lack GluA2 expression in non-transfected controls? Or controls that only receive mCherry plasmid transfections? Complete KO would require successful CRISPR targeting at both alleles, without donor insertion at either one. Either way, if complete KO in some cells is a possibility, this could be investigated more systematically (or at least discussed more thoroughly) since it may influence potential results.

To address the possibility of there being GluA2-lacking cells in non-transfected neurons, we have performed immunofluorescence on cultured mouse cortical neurons.

We found that 1.2% of neurons analyzed had no detectable levels of GluA2. Therefore, we can confirm the reviewers suspicious that a small fraction of neurons is, indeed, naturally GluA2-lacking. In comparison, we identified around 12.4% of cell with undetectable GluA2 (i.e., KO) with transfection of TKIT constructs for SEP-GluA2 KI (Figure 4—figure supplement 2D). Therefore, we think that undetectable GluA2 observed in some cells transfected with TKIT plasmids is an unfortunate, but not completely unexpected, result of gene knockout through double strand break generation without donor insertion. A possible explanation for the creation of these GluA2-KO neurons could also be incomplete co-transfection of all three plasmids (gRNAs/spCas9, donor DNA, mCherry). Cells lacking the donor DNA plasmid will be biased towards gene knockout. We further speculate that the discrepancy between the genomic PCR and immunofluorescence could result from a more complete co-transfection using electroporation and a bias towards easier detection of larger PCR products (i.e., the KI and WT fragments) using ethidium bromide integration for detection.

We have now included these data in the manuscript as Figure 4—figure supplement 3 and discussed this in the context of the GluA2 KO seen with TKIT (Results). Also see the Discussion. We also added a description of this methodology to the Materials and methods section of the manuscript.

2) Recent reports suggest that AAV vector genomes can integrate at DSBs created by Cas9 (PMIDs: 33087932, 31570731), and Cas9 can often create an opportunity for unintended insertions. Although PCR of genomic DNA (shown in Figure 4—figure supplement 2) detected both WT and KI alleles, it is not clear how often insertion of other sequences (with variable length) occurs, and no data at the level of DNA sequence is provided following AAV delivery methods. The fidelity of clean donor insertion could be examined by Sanger sequencing of genomic DNA clones to more completely define the results of insertion and genomic sequence changes around cut sites. This would also help to identify potential issues that may affect non-coding sequences (which, at the locations targeted here, are likely to be important regulatory elements).

We are grateful to the reviewers for raising this important point. We agree that this is a necessary consideration when coupling AAVs with CRISPR/Cas9 methods for genome editing (which is being widely adopted in the field). In fact, we now also speculate that 5’UTR integration of fragments of AAV could explain the lower efficiency in vivo with AAVs compared to in vitro with transfections. We have updated the text in the Results section to account for this possibility and include the recommended citations:

“Recent reports show that fragments of AAVs can integrate into double stranded DNA breaks induced by CRISPR/Cas9 (Hanlon et al., 2019; Wolter et al., 2020). […] Integration of AAV fragments into the 5’UTR of GluA1 or GluA2 could disrupt function, potentially contributing to the decreased efficiency observed in vivo (Figure 5H) compared to in vitro (Figure 2F).”

As the AAV approach is most useful for in vivo application, in future experiments (as part of a post publication Research Advance update following *eLife’s* new policy) we propose to inject the hippocampus of WT mice with a dual virus cocktail of AAV-spCas9 and AAV-TKIT-SEP-GluA2. The hippocampus will then be dissected, and genomic DNA amplify using a high-fidelity polymerase and a very long extension time (to account for variable sized insertions) flanking the KI sites. The PCR products will be sent for deep sequencing (as in Hanlon et al., 2019). This will enable us to calculate the percentage of reads that have AAV insertions.

3) I agree with the authors that tagging endogenous proteins is likely to preserve the normal physiological processes of transcription, mRNA processing, and protein localization, all of which will be specific to the target in question. Thus, the ability to use this system with proximity labeling proteomics approaches is a promising future direction. However, the author's claims about the superiority of this combination in the Discussion section should likely be tempered given that the tags used by these approaches were not incorporated or tested here.

We agree with the reviewers and have tempered this claim for future utility accordingly in our Discussion by adding the following sentence:

“However, the feasibility of coupling TKIT with proximity labelling remains to be tested and will likely depend upon the transfection or transduction efficiency, the KI efficiency for the target protein, as well as the absolute abundance of the target protein being tagged.”

4) Was the sequential sgRNA cloning and ligation process used for all genes targeted here? While it clearly works, this is somewhat laborious and may discourage others from adopting this technology. Since all TKIT targets will require two sgRNAs, the authors might consider using pX333 (PMID: 25337876; Addgene plasmid 64073) instead of pX330. This would enable more rapid sgRNA insertion (due to two unique restriction digest sites) instead of sequential amplification, gel purification, and ligation.

We agree with the reviewers and want to make this method as accessible to the community as possible. In fact, to simplify cloning of plasmids containing Two U6 + guides + cas9 we have designed a strategy using HiFi assembly to make these plasmids in a single cloning step. We have added this methodology to the Materials and methods section of the manuscript, which we hope will make it easier for investigators to use TKIT for their own projects. We also include the reviewer’s suggestion of using PX333.

However, for transparency, we will leave the details of how we originally made these constructs.

5) Regarding the FRAP experiments, could the authors expand their discussion on the difference between their measurements obtained on endogenous levels of receptors and those previously obtained on overexpressed receptors.

We thank the reviewers for this useful suggestion, which is at the core of why we think such CRISPR knock-in approaches are of such value to the field. Comparisons and citations of how our FRAP data fits within the literature of SEP-GluA1 overexpression is now summarized in the Results section of the manuscript:

“These findings are comparable to some (Heine et al., 2008; Kerr and Blanpied, 2012), but not all (Makino and Malinow, 2009), previous reports utilizing overexpression of SEP-GluA1. We speculate that this is critically dependent on the degree of overexpression, with very high levels of overexpression increasing the mobile pool of AMPARs beyond what is observed under more physiological conditions.”

6) In the cell specific approach, could the author provide feedback on Cas9 cell expression versus viral Cas9 expression.

In terms of the cell type-specificity, we expect the AAV-Cas9 approach to target all cells as Cas9 is expressed under the ubiquitous nEF promoter (hybrid EF1α /HTLV). In the transgenic approach, we expect much greater cell type-specificity for PV or CaMKII neurons, respectively. However, we have not validated this specificity explicitly, and it is not uncommon for Cre diver lines to have some leaky expression. Therefore, we cannot rule out the possibility of some off-target Cas9 expression as a result of the Cre driver line. We expect the levels of Cas9 will be more variable in the AAV-Cas9 approach than in the Cas9 knock-in mouse line, as some cells will take up more viral copies than others. We can foresee cell type-specificity being achieved with viral deliver of Cre-dependent Cas9 in different Cre-driver lines, which would save time and costly mouse breeding.

In addition, we speculate (but do not have the data to show) that the efficiency of TKIT could be higher using transgenic delivery of Cre, negating the need for dual virus uptake into the same cell, which can sometimes be problematic in neurons. We suggest leaving this speculation on efficiency out of the manuscript until we have solid data to support this claim. We propose adding these experiments as a post-publication Research Advance update, as per *eLife’s* policy.